# Laser-Welded Corrugated-Core Sandwich Composition—Numerical Modelling Strategy for Structural Analysis

Peter Nilsson [1], Seyed Rasoul Atashipour [2,3,*] and Mohammad Al-Emrani [4]

1    WSP Sweden, Fabriksgatan 1, SE-412 50 Gothenburg, Sweden; peter.c.nilsson@wsp.com
2    Department of Mechanical Engineering, Kettering University, 1700 University Ave, Flint, MI 48504, USA
3    Department of Mechanics and Maritime Sciences, Division of Dynamics, Chalmers University of Technology, SE-412 96 Gothenburg, Sweden
4    Department of Architecture and Civil Engineering, Chalmers University of Technology, SE-412 96 Gothenburg, Sweden; mohammad.al-emrani@chalmers.se
*    Correspondence: ratashipour@kettering.edu or rasoul.atashipour@chalmers.se; Tel.: +1-810-762-9500 or +46-31-772-1500

**Abstract:** Laser-welding technology has recently enabled the production of corrugated-core steel sandwich panels (CCSSPs) as an innovative large-scale, lightweight structural solution in maritime and infrastructure applications. Detailed numerical analyses, specifically weld-region stress prediction in the presence of transverse patch loading and supports, are computationally challenging and time-consuming for their optimal design. This paper introduces an efficient, simplified combined sub-modelling approach for accurately predicting the detailed structural response of welded corrugated-core steel panels. The approach rests on the homogenisation of the three-dimensional (3D) panel into a two-dimensional equivalent orthotropic single layer (EOSL), where the effect of transverse compressive loads and local support conditions are captured separately via different 2D and 3D sub-modelling techniques, together with a model introduced for calculation of the weld region's equivalent spring stiffnesses. A laser-welded corrugated-core steel sandwich panel (CCSSP), as a future generation of the steel bridge deck, was examined using different modelling approaches. It was shown that the proposed combined sub-modelling approach can accurately predict stresses and displacements in all the constituent members of the cross-section, including the welds, in a reasonable calculation time when compared with a 3D reference model, unlike the conventional homogenisation approaches.

**Keywords:** corrugated-core steel sandwich panel (CCSSPs); compressive load effect; combined sub-modelling approach; equivalent orthotropic single layer (EOSL); weld-region-equivalent spring stiffness

## 1. Introduction

Sandwich panels with a corrugated core have been used in many branches of modern technology, namely maritime and shipbuilding (e.g., see [1–3]), aviation and aerospace [4–6], automotive [7–9], and building and bridge structures [10–13], where a lightweight load-bearing panel is the main target of the application. This is due to the high potential and flexibility of the topology for an optimal design of a high-strength lightweight panel, whilst the shape enables production.

In recent years, laser-welding technology has enabled the production of corrugated-core panels made of steel as an innovative lightweight, large-scale sandwich structural solution in maritime and infrastructure applications (Figure 1).

Corrugated-core steel sandwich panels (CCSSPs) structurally behave in a complex and orthotropic manner by nature. Accurately predicting load effects in the constituent plates and interface connections/welds of the cross-section is demanding. The level of complexity increases further when considering regions at localised loads or supports. One

possibility is to perform a three-dimensional (3D) numerical analysis with shell or solid elements (e.g., for a bridge with a CCSSP deck [12,13]). However, these models need to be detailed, and a system of equations with a large number of unknowns needs to be solved. That requires significantly high computational effort (see, e.g., [14]), which is inappropriate for design and optimisation. Therefore, a simplified model is sought to be strategically implemented for structural analysis in such a way that, depending on the type of required structural results, the accuracy of the gained numerical results shall not be affected by the simplifications. This is indeed of high importance when the analysis serves a design or optimisation purpose.

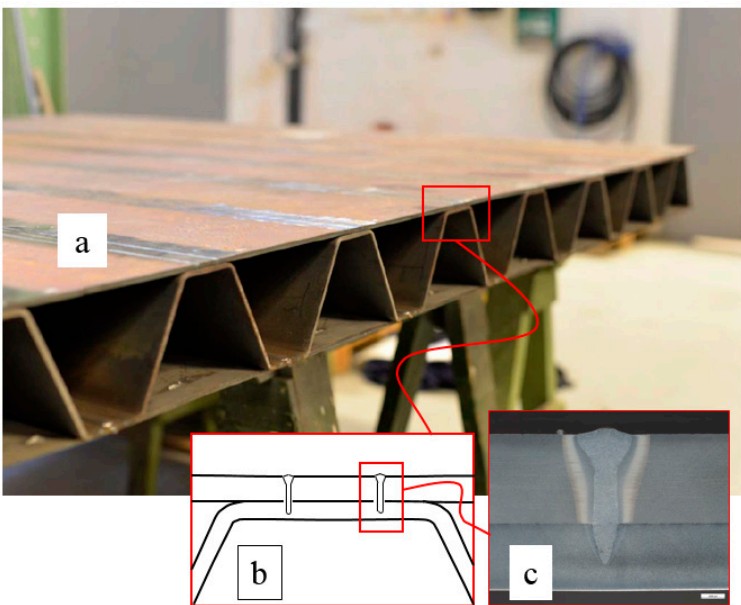

**Figure 1.** Corrugated-core steel sandwich panel: (**a**) the laser-welded panel, (**b**) core-to-face joint, and (**c**) weld region [13].

Recently, Hammarberg et al. [15] carried out a structural analysis and stiffness evaluation for bidirectional corrugated-core steel sandwich panels. They employed a homogenisation approach by describing the combination of the complex core and the face plates by averaging equivalent finite elements to reduce the number of elements and enable structural optimisation. Using the homogenization technique, Romanoff and his co-workers [16] performed a series of research on computational analyses and optimisation of laser-welded steel sandwich beams and panels with the web-core configuration. For methods of homogenization of corrugated panels, readers are referred to the work of Biancolini [17], Kress and Winkler [18], and Nilsson et al. [12], where both analytical and numerical approaches are presented.

Homogenisation approaches are valid only for the global effects and cannot capture the local effects. Therefore, significant research efforts have contributed to the development of structural analysis methods for large-scale sandwich panels; e.g., see Klosterman [19], Romanoff and Varsta [16,20], and Karttunen et al. [21,22]. However, very modest attention has been directed towards laser-welded CCSSPs. One method that was proposed for web-core sandwich panels by Romanoff et al. [23] utilises, in addition to the analysis of the homogenised panel, a separate and decoupled local analysis for the contribution of directly applied load (DAL). In this paper, this methodology is denoted as the EOSL approach.

Another modelling approach used for structural analysis calculations of CCSSPs is a deformation-driven sub-modelling approach using finite element analysis (FEA). The main intention of this well-established modelling technique is to obtain detailed results in a specific region, whereas the main part of the structure is modelled with a lower degree of detailing. First, a global model is run, and the deformation output of the global model is used as boundary conditions (BCs) for a local, more detailed model. This methodology

is commonly used for design purposes in several applications; see, e.g., Aygul et al. [24], where this methodology is used for fatigue design of welded details.

In a CCSSP with discretised connections/weld lines between constituent elements, the connections need to be modelled by springs with a finite stiffness when a 3D shell modelling technique is employed, as can be the case in the sub-modelling approach. Only a few research studies have dealt with determining the aforementioned stiffness for sandwich plates; e.g., the rotational stiffness of laser stake welds, both considering web-core sandwich panels (Romanoff et al. [25]) and for CCSSPs (Nilsson et al. [26]). This has been done to incorporate the proper stiffness of the weld region in an analysis that uses structural elements, i.e., beams or shells. However, regarding CCSSPs, the 2D approach in [26] has not been verified for 3D analyses. Furthermore, in CCSSPs, the core-to-face joint, which consists of a weld pair (Figure 1), is a stiff region that undergoes small deformations under loading. The effect of weld-region deformations, other than the rotations (i.e., in the vertical and horizontal directions), has not been investigated.

In recent research studies, developments of improved plate and beam theories for discrete core sandwich structures have been presented. One of these methodologies is the coupled stress method; see Goncalves et al. [27]. Another methodology that has shown great promise for web-core sandwich beams and panels is the micropolar theory (see Karttunen [22]). The constitutive relationships for web-core sandwich beams were presented by Karttunen et al. [28] and later for web-core sandwich panels by Karttunen et al. [22]. This methodology rests on energy equilibrium between a micro- and a macro-scale analysis compared to the enforced displacements used in the sub-modelling approach, giving it a significant benefit. However, this methodology does not apply to CCSSPs with dual weld lines and interacting with a supporting structure.

The present paper aims to introduce an efficient combined sub-modelling approach that enables structural design and optimisation of CCSSPs as parts of larger structures (see Figure 2) via accurate predication of displacements and stresses at both global and local scales, e.g., at the welding zones and the constituents' sub-elements with a reasonable amount of computational effort. Different modelling approaches are evaluated for the CCSSP in interaction with a supporting structure, and different scenarios for supporting effects (e.g., transverse girders) as well as localised transverse loads are considered, and thereby, an appropriate combined sub-modelling approach that can be used for the design of CCSSPs with dual weld lines is identified and introduced. This study also highlights some aspects specific for CCSSPs with dual weld lines in terms of their load-carrying behaviour. In this perspective, special focus is put forward on the impact of deformations in the core-to-face joint and the weld region.

This manuscript is organised as follows. Section 2 presents different modelling techniques as parts of the introduced combined sub-modelling approach, including three categories: (1) 2D structural models: 2D-1: frame model for panel-level bending and weld-region analyses, 2D-2: a constrained frame model to capture the effect of directly applied compressive transversal loads (DAL), and 2D-3: solid element model for weld-region analyses; (2) 3D structural models: 3D-1: global model of a large-scale CCSSP structure entirely made of an equivalent orthotropic single-layer (EOSL), 3D-2: second-level large sub-model based on 3D shell and importing deformations from 3D-1 as boundary conditions, 3D-3a: third-level small sub-model with 3D shell and uses deformation from 3D-2 as boundary conditions to capture various supporting scenarios, and 3D-3b: analogous model to 3D-3a but based on solid elements; and (3) Models for weld-region-equivalent spring stiffness: WLRS: rotational stiffness about the weld line, WLAS: axial stiffness of the weld line, and WLSS: weld-shear stiffness perpendicular to the weld line. Section 3 presents the numerical comparative results and discussion. The conclusions are drawn in Section 4.

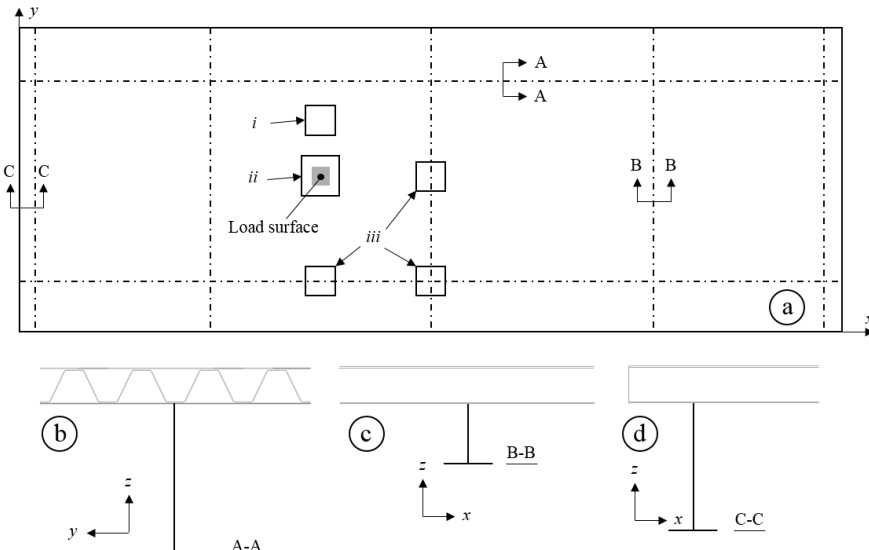

**Figure 2.** Investigated regions in CCSSPs corresponding to different transverse load and supporting scenarios: (i) area subjected to panel-level sectional forces alone, (ii) loaded region, and (iii) at the connection to supporting structure. (**a**) View from above, (**b**) section at main girder, (**c**) section at intermediate cross-beam, and (**d**) section at end cross-beam.

## 2. Combined Sub-Modelling Approach

In order for a model of a large-scale steel sandwich panel connected to a supporting structure to be appropriate for design and optimisation, it needs to be able to accurately capture the following: (1) average deformations of the panel, (2) stresses in all constituent plates of the cross-section, and (3) stresses in the welds. All of these demands shall be fulfilled in several important regions in the panel and under various loading situations. These regions can be distinguished by transverse loading situation and the location at (i) positions subjected to panel-level sectional forces alone (away from loads and supporting structure); (ii) positions subjected to DAL; and (iii) connections to the supporting structure (see Figure 2). Even though several previous studies have contributed to the development of plate analysis methods for laser-welded sandwich panels, very limited research has been focused on panels that are connected to a supporting structure, where they regard web-core sandwich plates (Romanoff et al. [23] and Romanoff [29]).

In this section, different modelling techniques as parts of the combined approach to be presented here are described in detail. It should be clarified that some of the following models were also used in the verification process of the investigated structural analysis approach or strictly to visualise a certain structural behaviour that needs further investigation.

In all the modelling parts, a general corrugated-core panel cross-section, according to Figure 3, was utilized, where $h$ is the vertical distance between middle surfaces of face sheets, $2p$ is the corrugation pitch, $\theta$ is the angle between face sheets and straight diagonal portion of corrugation leg, and $R_1$ and $R_2$ are the corrugation bend radius of the bottom and upper corners, respectively. In addition, $t_1$ and $t_2$, are, respectively, the bottom- and upper-face plate's thickness, and $t_c$ is that of the corrugated core sheet; $t_w$ is the weld width, and $h_g$ is the gap between the core and face plates. Also, $d_w$ is the distance between the weld lines, $\eta$ indicates the length of the straight part of the corrugated plate from a weld line to its crest/trough, and $f = 2\eta + d_w$ (see Figure 3).

All models in 2D space are adopted with the plane stress conditions. Moreover, for the models having "equivalent orthotropic single layers" (EOSLs), the equivalent cross-sectional stiffness property components need to be determined, including in-plane axial stiffness in the stiff direction ($E_x$), in-plane axial stiffness in the weak direction ($E_y$), in-plane shear stiffness ($G_{xy}$), bending stiffness in the stiff direction ($D_x$), bending stiffness in the weak direction ($D_y$), twisting stiffness ($D_{xy}$), transverse shear stiffness in the stiff direction

($D_{Qx}$), and transverse shear stiffness in the weak direction ($D_{Qy}$). All the mentioned equivalent elastic constants for the EOSLs are determined according to the approach of Libove and Hubka [30] except for the transverse shear stiffness in the weak direction, $D_{Qy}$, and the axial stiffness in the $y$-direction, $E_y$. The shear stiffness component $D_{Qy}$ is calculated according to Nilsson et al. [12] to account for the dual-weld connection between the core and face plates. Regarding the axial stiffness $E_y$, as this aspect was disregarded in [30], and a numerical analysis was conducted in order to investigate the impact of core stretching.

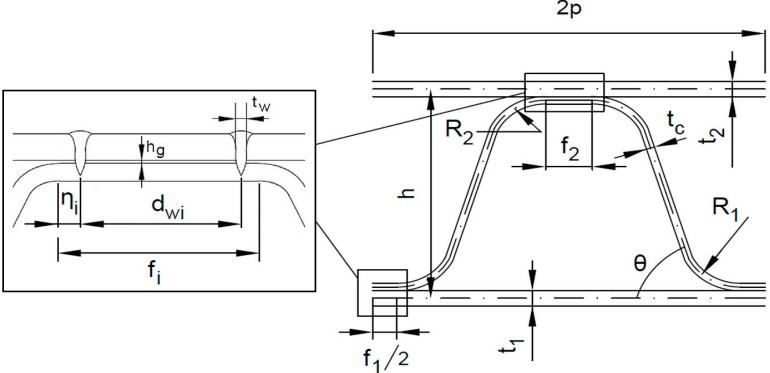

**Figure 3.** Cross-sectional geometry of corrugated core steel sandwich panels (CCSSPs) with dual weld lines [12].

*2.1. Two-Dimensional Structural Models*

2.1.1. Frame Model for Panel-Level Bending and Weld-Region Analyses (2D-1)

A 2D beam model in 2D space was considered first (2D-1). The beam is a transverse section of a larger structure (see Figure 2) and is loaded in four-point bending, as it is shown in Figure 4. Supports are located at the positions of the main supports according to Figure 2. As mentioned previously, the beams are of unit width (i.e., plane stress conditions) along the $x$-direction.

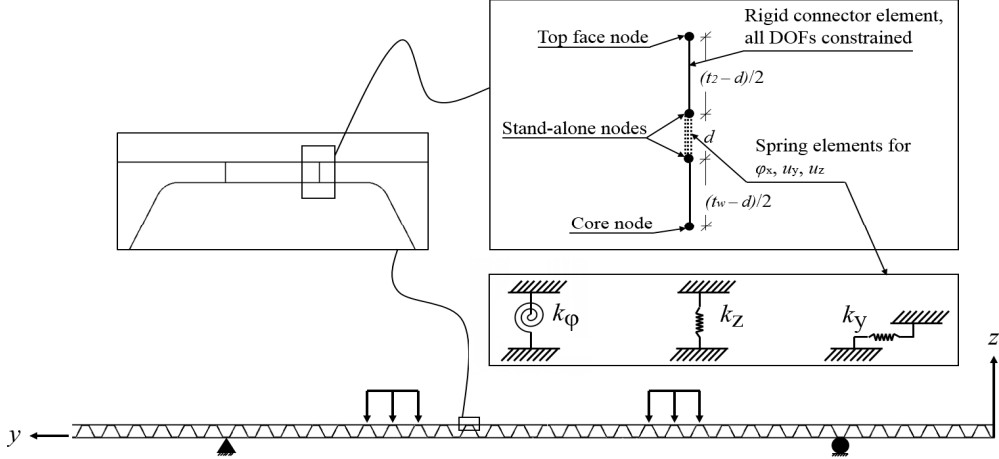

**Figure 4.** Model 2D-1: Frame model of a transverse cut of a large CCSSP structure in four-point bending and schematic of the weld-region model with springs.

This model was used for two purposes: the first was to determine the stress components in the constituent members of the cross-section under pure bending action (which is valid between the two loads) in conjunction with the EOSL approach. The secondly was to investigate the impact of the weld-region deformations on the weld stresses. The model contains three spring elements representing the deformation of the weld region, as illustrated in Figure 4, where $d$ indicates the distance between the stand-alone nodes.

### 2.1.2. Constrained Frame Model for Effect of DAL (2D-2)

In order to investigate the DAL (i.e., the local load effects) contribution in the EOSL approach in the form of stresses in the constituent plates of the cross-section, a special constrained frame model was used here (2D-2). The model is analogous to 2D-1, however, with the global action constrained at each bottom core-to-face joint to extract the local load effects, as shown in Figure 5.

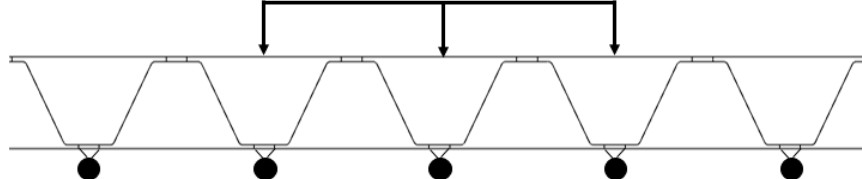

**Figure 5.** Model 2D-2: Frame model for decoupled directly applied load (DAL) contribution in the equivalent orthotropic single-layer (EOSL) approach.

### 2.1.3. Solid Element Model for Weld-Region Analyses (2D-3)

This model is 2D solid (called here 2D-3); see Figure 6. This model was used as a reference for investigation of the impact of the weld-region deformations on the weld stresses.

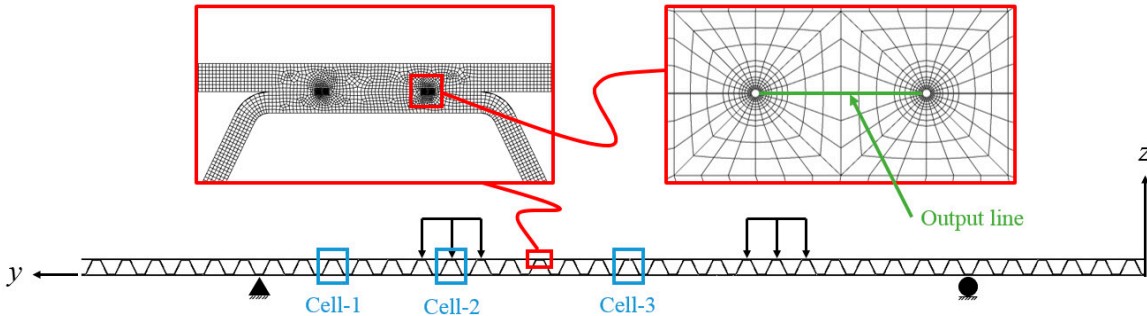

**Figure 6.** Principle sketch of the 2D solid model (2D-3).

To calculate nominal stresses in the welds (that are compared to 2D-1), the stress distribution along the output line (see Figure 6) was integrated, and the sectional forces in the welds and the corresponding linear stress distributions were calculated. From the linear stress distribution, the nominal normal weld stress from bending $\sigma_{zM}$ and from axial force $\sigma_{zN}$ was calculated as illustrated in Figure 7. In this figure, $y_w$ is the coordinate along the weld (output line in Figure 6).

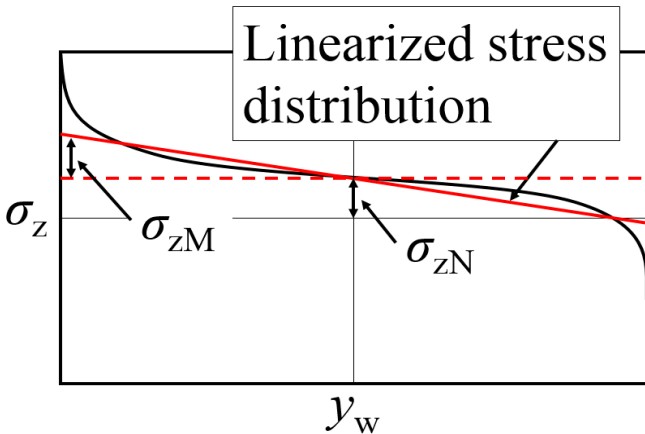

**Figure 7.** Calculation of nominal stresses from the stress distribution of the solid model.

### 2.2. Three-Dimensional Structural Models

In this section, all models confined in the 3D space are described in detail as follows:

- 3D-1: Global model of a large-scale CCSSP structure, which is entirely made of EOSL (Figure 8a);
- 3D-2: Level 2 large sub-model with 3D shell, which uses deformation from 3D-1 as boundary conditions (Figure 8b);
- 3D-3a: Level 3 small sub-model with 3D shell, which uses deformation from 3D-2 as boundary conditions—various supporting scenarios are considered (Figure 8c);
- 3D-3b: Analogous to 3D-3a but with solid elements (Figure 8d).

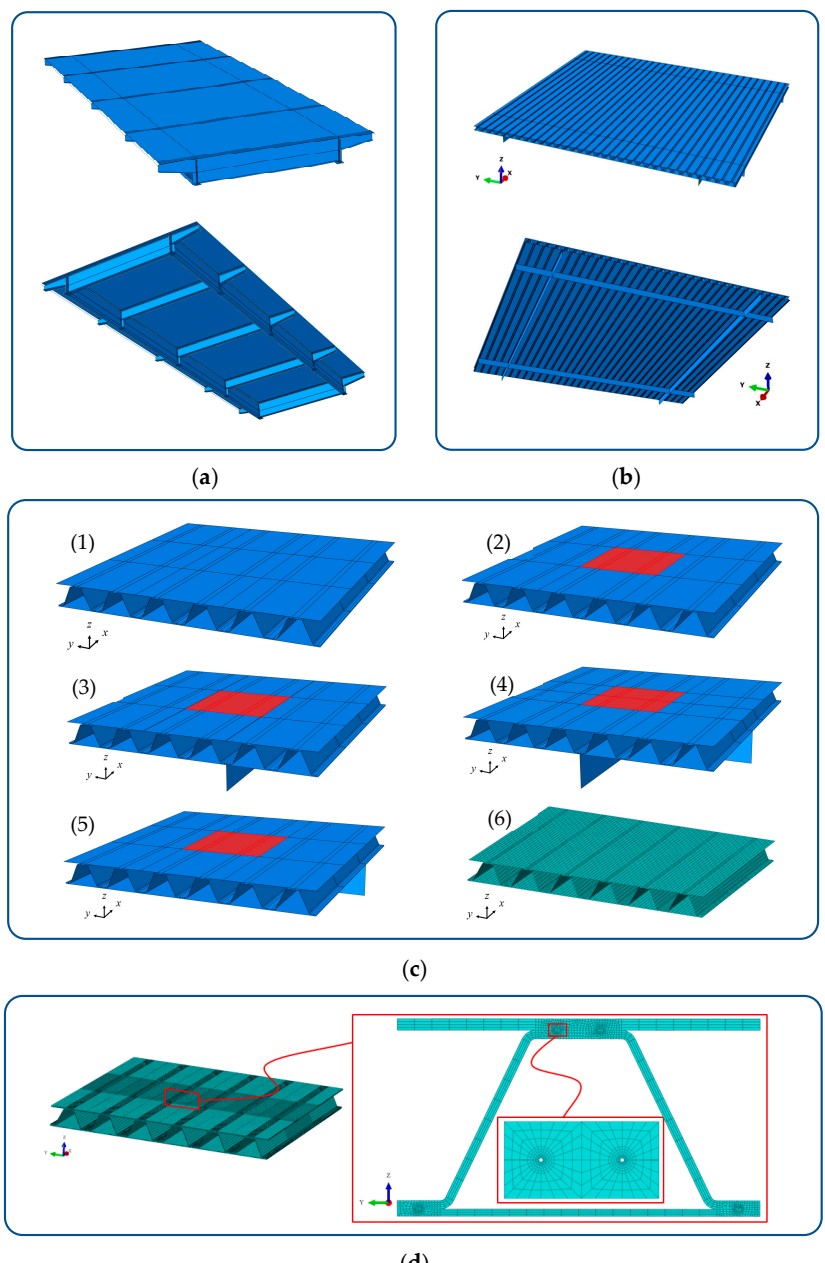

**Figure 8.** Illustrative overview of different considered modelling approaches (both top and bottom views): (**a**) global model (3D-1): orthotropic equivalent single-layer (EOSL) deck and rest of the structure as isotropic shells; (**b**) level 2 sub-model (3D-2); (**c**) level 3 shell-element sub-models (3D-3a) at region: (1) i, (2) ii, (3) iii-A, (4) iii-B, (5) iii-C, and (6) typical mesh density (the red region indicates applied load location); (**d**) level 3 solid-element sub-model (3D-3b).

Clearly, the EOSL approach is the most time-efficient and depends only on the 3D model 3D-1 together with the simplistic 2D models. In the sub-modelling approach, the models 3D-1, 3D-2, and 3D-3a were used, and the validity of the results was examined by the sub-model 3D-3b. Each higher, more detailed level of analysis used the deformations from the previous lower-level model as driving BCs. Thus, the sub-modelling approach utilised a three-level multiscale model.

As stated earlier, three different load/BC situations are considered in this paper. Situation iii includes three different cases yielding five regions in total that were investigated, named i, ii, iii-A, iii-B, and iii-C. The regions together with their respective loading situations are shown in Figure 9 (also see Figure 2 for orientation). It should be noted that i represents global load effects alone, ii describes global load effects together with local load effects from DAL, and iii-A to C address global load effects, effects from DAL, and local support conditions.

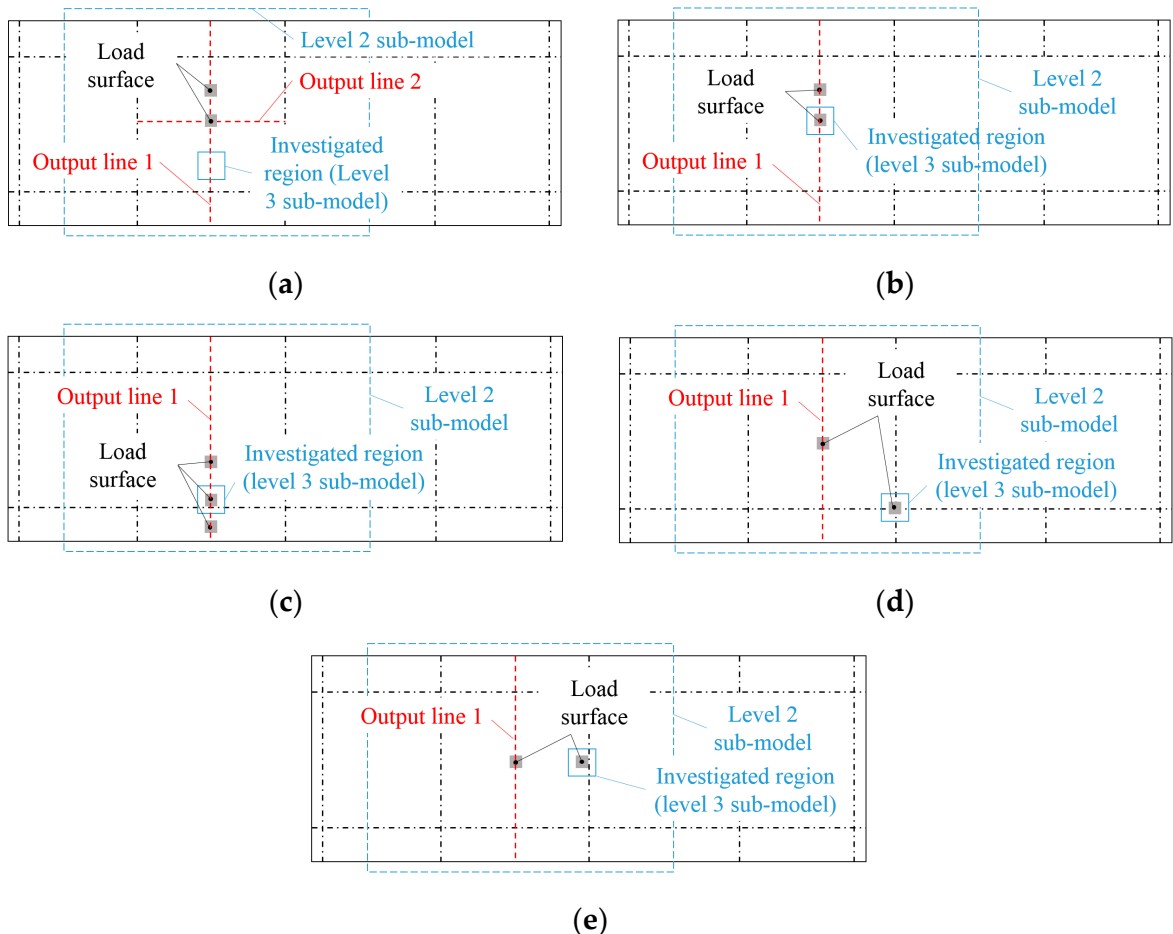

**Figure 9.** Loads and position of level 2 and level 3 sub-models, view from above: (**a**) region i, (**b**) region ii, (**c**) region iii-A, (**d**) region iii-B, and (**e**) region iii-C.

Figure 8a shows an overview of the model implemented on a bridge deck example, and Figure 10 gives all relevant geometric parameters of the global structure.

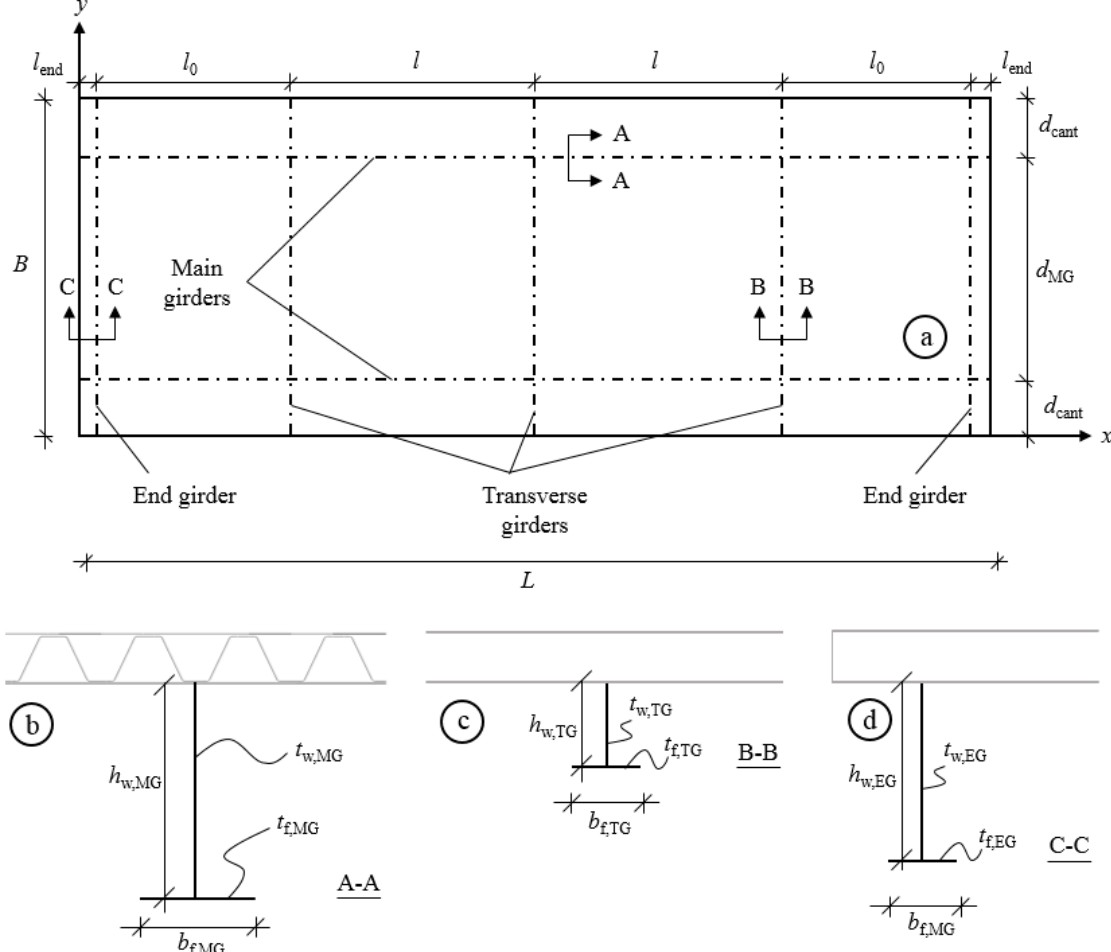

**Figure 10.** Geometric notations for global model: (**a**) view from above, (**b**) section A-A, (**c**) section B-B, and (**d**) section C-C.

2.2.1. Global Model with EOSL Orthotropic Panels (3D-1)

Based on this model (3D-1), a full, large CCSSP structure was considered as EOSL shear-deformable plates. This approach consisted of the four steps shown in a schematic image for a CCSSP in Figure 11.

The first step of this methodology was to (a) derive from a unit cell the equivalent stiffness properties in all relevant directions and secondly to (b) solve the plate equations for the EOSL using numerical or analytical methods. The plate equations follow the first-order shear-deformable Reissner/Mindlin kinematic relationships (see, e.g., Libove and Batdorf [31]) with or without the thick-face effects (see, e.g., Romanoff and Varsta [20]). Within the transformation from 3D to 2D, the information concerning the discrete nature of the sandwich panel is lost. This leads to the third step, (c) where the discrete structure is reconsidered. The panel-level sectional forces calculated in step b are used to load the unit cell, and then, the load effects in the constituent members of the cross-section are determined via the definition of the corresponding equivalent stiffness property. In this step, a constant force or bending moment acts on the unit cell, which is an approximate inherent in the approach. In the last step, (d) the effect of DAL or local support conditions is considered by a separate decoupled analysis either in 2D or 3D, and the results are superimposed on those from the panel-level sectional forces.

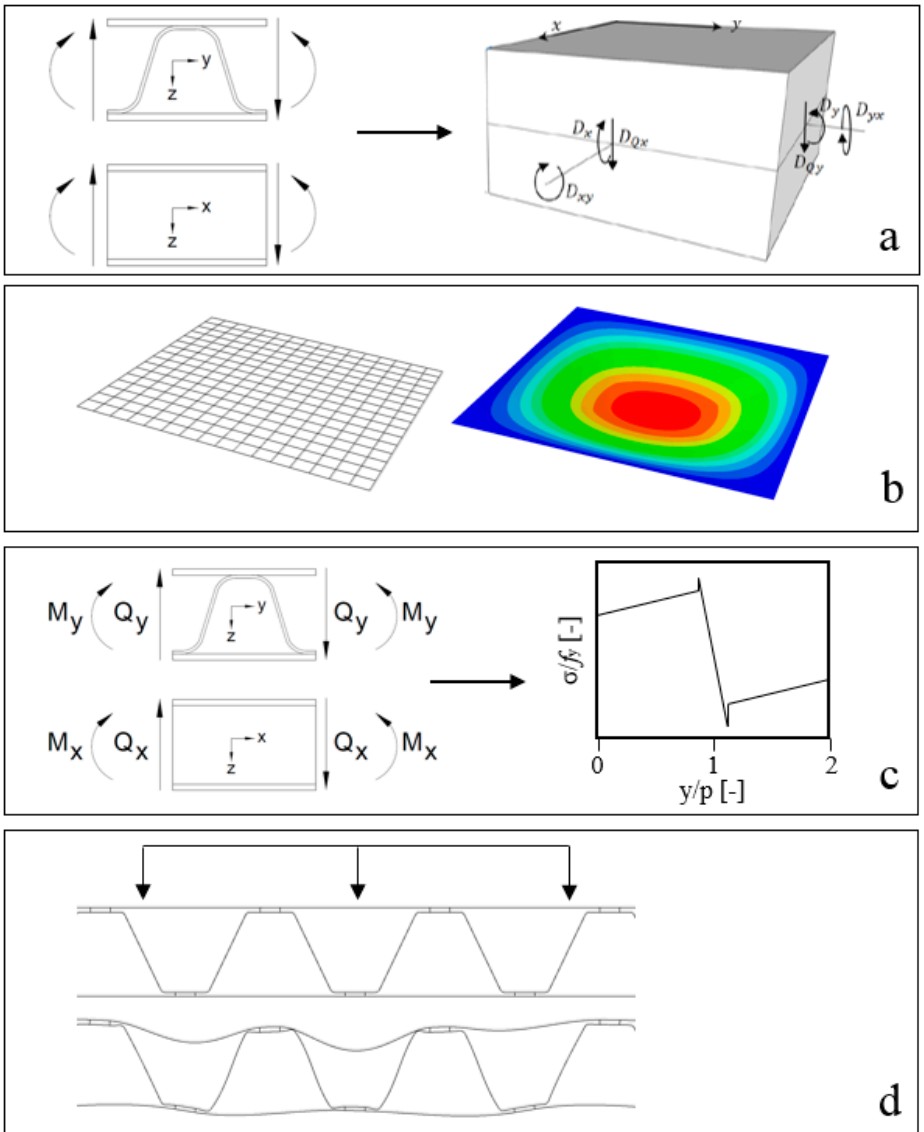

**Figure 11.** Schematic view of EOSL approach: (**a**) derivation of equivalent stiffness properties, (**b**) analysis of equivalent orthotropic single layer (EOSL), (**c**) panel-level sectional forces causing local plate deformations yielding local stress distribution in the constituent members of the cross-section, and (**d**) effect of directly applied loads (DAL).

### 2.2.2. Large Sub-Model with 3D Shells (3D-2)

The level 2 sub-model (3D-2) is a large shell model of the CCSSP, and it includes a part of the supporting structure as shown in Figure 8b. Model 3D-2 uses the deformations from the model 3D-1 as boundary conditions. The location of this sub-model with respect to the global model 3D-1 is shown in Figure 9 for five different regions, covering different possible scenarios with regard to load and support conditions.

When a large CCSSP structure is modelled in the form of the mentioned 3D shell configuration, the weld-region deformation needs to be accounted for. This is performed analogous to the model 2D-1 for each face-to-core node pair along the weld line, as shown in Figure 12.

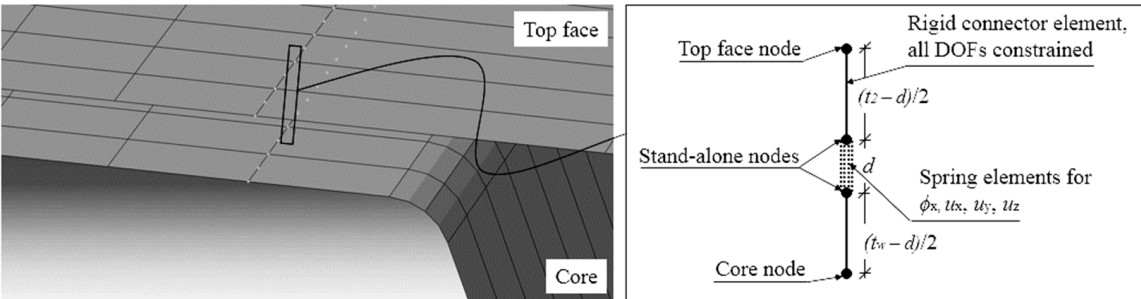

**Figure 12.** Modelling of weld region in 3D shell models.

It should be clarified that the spring related to the deformation along the extrusion direction (parallel to the weld line) is considered infinitely stiff.

### 2.2.3. Small Sub-Model with 3D Shells (3D-3a)

The level 3 sub-model 3D-3a is a shell model equivalent to 3D-2 but of a smaller region and with a higher mesh density (Figure 8c). This model adopts the deformations from 3D-2 as BCs. Also, the weld region was modelled for this model according to Figure 12.

### 2.2.4. Small Sub-Model with 3D Solids (3D-3b)

The level 3 sub-model 3D-3b is a solid model analogous to 3D-3a, as shown in Figure 8d. This model typically contains a large number of degrees of freedom yielding time-consuming analyses. This model type was used as a reference for comparison purposes to investigate the validity and accuracy level of the other models. The nominal stresses in this solid model were calculated analogous to what described in Section 2.1.1.

It is noteworthy that since EOSL analysis can only capture the average mid-plane deformations and not the detailed deformed shape of all the constituent members, a challenging point would therefore be to connect an EOSL model like 3D-1 to a geometrically modelled sub-model 3D-3a or 3D-3b, as the cut in the *yz*-plane (i.e., along the in-plane direction perpendicular to the corrugation extrusion direction) forces a deformed shape in the sub-model boundaries that is unnatural for the panel. For this reason, the *yz* cut must be placed sufficiently far from the investigated region. In addition, this model only needs to be converged with respect to deformations and not stresses in order to derive the more detailed level 3 sub-model.

### 2.3. Models for Weld-Region-Equivalent Spring Stiffness

The weld lines are proposed here to be modelled in the form of combined rotational, axial, and shear springs with equivalent stiffness values determined as described in the following. The springiness along the weld line is relatively rigid and therefore can be considered infinitely stiff.

In order to calculate the magnitudes of the rotational and translational spring stiffness components, 2D solid models were used for the configuration shown in Figure 13a. In this figure, $t_1$ and $t_2$ are two adjacent welded CCSSP's constituents plates; i.e., upper face-core or core-bottom face plates. $L$ is the model length, $h_g$ is the core-to-face gap, and $t_w$ is the weld width. It is noted that for a general CCSSP in which the face-plate thicknesses are not equal, the equivalent spring stiffness will obviously be different for the welds in the core-to-face plate joint at the top and bottom of the panel. The exterior cross-sectional edges of the weld deformation models, $e_i$, were set to be rigid, and the load or deformations were applied in the centric node, $n_i$ (Figure 13b).

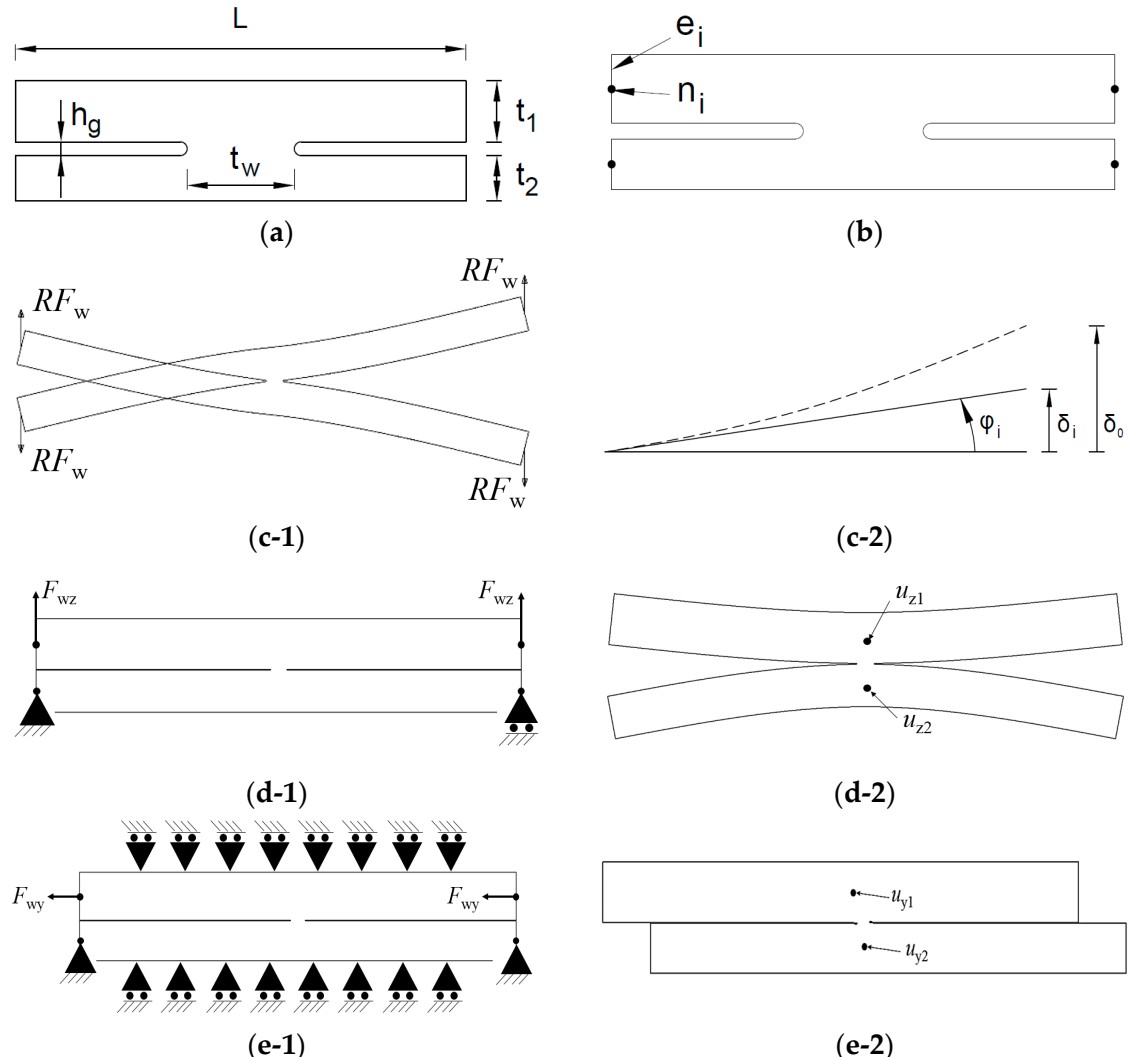

**Figure 13.** Model for calculation of weld-region-equivalent spring stiffness values: (**a**) geometrical configuration of the model; (**b**) rigid exterior edges and load application points; (**c**) rotational spring stiffness calculation: 1—reaction forces from FEA, 2—rotational deformations; (**d**) axial spring stiffness calculation: 1—loads and boundary conditions, 2—axial deformations; (**e**) shear spring stiffness calculation: 1—loads and boundary conditions, 2—shear deformations.

### 2.3.1. Rotational Stiffness about the Weld Line (WLRS)

This stiffness magnitude can be calculated directly according to the closed-form equation derived by Nilsson et al. [26]. However, as that solution is derived using the results from a large number of numerical analyses that are evaluated statistically, a minor error will be inherent. Therefore, to eliminate this error, the rotational spring stiffness components can also be calculated explicitly for any studied case according to the model approach presented in Figure 13c-1,c-2. Moreover, the methodology used to calculate $k_\varphi$ here is analogous to the calculations described in [26]. A unit deformation, $\delta_0$, was applied to the loading points, $e_i$, and the reaction force $F_w$ was calculated via the FEA; see Figure 13c-1. From $F_w$, the bending deflection $\delta_b$ can be determined using a cantilever beam following Timoshenko kinematics (Figure 13c-2). In turn, $\delta_i = \delta_0 - \delta_b$ was used to describe the rotational deformation of the weld region, and $\varphi_i$ and $k_\varphi$ were determined by the relationship between the moment acting on the weld (determined by $F_w$ and $L$) and the rotation angle $\varphi_i$. For more details regarding the calculations of the rotational spring stiffness, see [26].



2.3.2. Axial Stiffness of the Weld Line (WLAS)

In order to estimate the axial stiffness of the weld region $k_z$, a unit load $F_{wz}$ was applied to the top plate-loading points. whereas the bottom loading points were constrained; see Figure 13d-1. The vertical deformations at the mid-plate position, centrically over the weld $u_{z1}$ and $u_{z2}$, were calculated via FEA for each plate (Figure 13d-2). Next, the axial deformation of the weld region, $\Delta_z$, can be calculated as $\Delta_z = u_{z1} - u_{z2}$. The vertical spring stiffness is defined by $k_z = 2F_{wz}/\Delta_z$.

2.3.3. Weld Shear Stiffness Perpendicular to the Weld Line (WLSS)

Regarding the weld-region deformations in the $y$-direction (the shear deformations of the weld), a structural model according to Figure 13d-1 was used for the derivation of the spring stiffness components. The applied load $F_{wy}$ caused the mid-plane nodes centric o the weld to deform for the upper and lower plate, $u_{y1}$ and $u_{y2}$, respectively, and the difference between the two points constituted the shear deformation of the weld region, $\Delta_y = u_{y1} - u_{y2}$; see Figure 13d-2.

**3. Numerical Results and Discussion**

Based on the introduced modelling approaches in the previous section, a comprehensive comparative numerical results and discussion is conducted here to examine and highlight the effectiveness and capability of different modelling approaches with regard to different demanding structural responses and to identify their relative deficiencies and drawbacks. A clear picture of an appropriate choice of modelling approach with regard to a required structural result is demonstrated to the readers, aiming at facilitating the structural engineers' selection when dealing with large CCSSP structures. The influence of various local and global loads and supporting effects is also taken into account.

A large-scale CCSSP as bridge deck structure was considered for the numerical results. The geometric properties and dimensions of the corrugated-core sandwich panel were assumed as presented in Table 1. Moreover, a weld width of $t_w = 2$ mm and a gap between the core and face plates of $h_g = 10$ μm was assumed.

**Table 1.** Cross-sectional properties for studied case deck.

| $t_1$ (mm) | $t_c$ (mm) | $t_2$ (mm) | h (mm) | θ (°) | $f_{1,2}$ (mm) | $d_{w1,2}$ (mm) | 2p (mm) | $R_{1,2}$ (mm) |
|---|---|---|---|---|---|---|---|---|
| 5 | 6 | 8 | 132 | 65 | 60 | 30 | 253 | 7.4 |

Linear elastic isotropic properties were considered for the steel constituent material with the Young's modulus $E = 210$ GPa and the Poisson's ratio $\nu = 0.3$. Furthermore, geometrically linear relationships—no contact action between core and face plates and no second-order effects—were assumed.

For the category of the 2D models, plane stress conditions were adopted, and unit width was considered. For all models having EOSLs, the equivalent orthotropic stiffness properties were calculated based on the approaches of Libove and Hubka [30] and Nilsson et al. [12] for the considered geometrical configuration (Table 1) and are presented in Table 2. As mentioned previously, all elastic constants were calculated according to [30] except for the transverse shear stiffness in the weak direction, which was calculated according to Nilsson et al. [12], and the axial stiffness $E_y$, which was computed according to a conducted numerical analysis in this section, as this stiffness property was disregarded in [30].

**Table 2.** Equivalent stiffness properties considering plane stress condition for EOSL (calculated based on the developed formulation in Libove and Hubka [30] and Nilsson et al. [12]).

| $E_x$ (N/m) | $E_y$ (N/m) | $G_{xy}$ (N/m) | $D_x$ (Nm) | $D_y$ (Nm) | $D_{xy}$ (Nm) | $D_{Qx}$ (Nm) | $D_{Qy}$ (Nm) |
|---|---|---|---|---|---|---|---|
| $4.72 \times 10^9$ | $3.05 \times 10^9$ | $1.36 \times 10^9$ | $15.5 \times 10^6$ | $11.5 \times 10^6$ | $8.79 \times 10^6$ | $334 \times 10^6$ | $12.7 \times 10^6$ |

All beam elements used for the category of 2D beam models are three-node shear-deformable, i.e., following Timoshenko kinematics, with second-order shape functions. For the 2D shell models, the eight-node shear-deformable element type with second-order shape functions was employed, and the solid models were conducted using twenty-node solid elements.

To demonstrate the capability of different relevant models/sub-models for capturing both the local and global transverse load effects, a transverse patch load distributed over $500 \times 500$ mm area and of magnitude of 135 kN was assumed. It should be pointed out that the considered transverse load configuration corresponds to a wheel pressure from the design vehicle load model 1 in CEN [32], with the Swedish national parameter of (0.9). Detailed geometric dimensions of the large CCSSP deck structure example (including short and large main girder transverse supports) in this section are given in Table 3.

**Table 3.** Geometric dimensions of the assumed large-scale CCSSP deck structure.

| $L$ (m) | $B$ (m) | $l$ (m) | $l_0$ (m) | $l_{end}$ (m) | $d_{cant}$ (m) | $d_{MG}$ (m) | $h_{w,MG}$ (mm) | $t_{w,MG}$ (mm) | $t_{f,MG}$ (mm) |
|---|---|---|---|---|---|---|---|---|---|
| 20.4 | 9.11 | 5.56 | 4.44 | 0.200 | 1.52 | 6.08 | 1000 | 8 | 20 |

| $h_{w,TG}$ (mm) | $b_{f,TG}$ (mm) | $t_{w,TG}$ (mm) | $t_{f,TG}$ (mm) | $h_{w,EG}$ (mm) | $b_{f,EG}$ (mm) | $t_{w,EG}$ (mm) | $t_{f,EG}$ (mm) | $b_{f,MG}$ (mm) | |
|---|---|---|---|---|---|---|---|---|---|
| 500 | 200 | 6 | 20 | 900 | 200 | 6 | 20 | 300 | |

The numerical values for all spring stiffness components calculated according to the models introduced in Section 2.3 for the assumed geometric configuration (see Table 1) are shown in Table 4, where indices 1 and 2 refer to the top and bottom core-to-face joints, respectively.

**Table 4.** Numerical values of weld-region spring stiffness components.

| $k_{\phi1}$ (kNm/(rad·m)) | $k_{\phi2}$ (kNm/(rad·m)) | $k_{z1}$ (MN/m) | $k_{z2}$ (MN/m) | $k_{y1}$ (MN/m) | $k_{y2}$ (MN/m) |
|---|---|---|---|---|---|
| 93.04 | 92.58 | 20.30 | 13.50 | 9.22 | 8.34 |

It is noteworthy that convergence was reached for all numerical models in such a way that increasing the size of the model did not affect the results of investigated regions for any cases. In the following, the numerical comparative results are presented and discussed.

### 3.1. Global Deflection Predictions

In this section, the average deflections calculated by the global model with an EOSL deck (3D-1) are compared to the deflections from the level 2 sub-models (3D-2). As an output component, a normalized deflection $\bar{u}_z = u_z / |u_{z,\max,3D\text{-}2}|$ was used, where $u_z$ was the vertical deformation (deflection) for the respective model, and $u_{z,\max,3D\text{-}2}$ was the maximum deflection within the path of the 3D-2 model (output line 1 in Figure 9). Figure 14 shows comparison of $\bar{u}_z$ versus $\bar{y} = y/B$, where $B$ is the bridge deck width, between the EOSL model and the 3D shell element model 3D-2, for the five regions i to iii-C. The bottom face deflections of 3D-2 are compared to the deflections of the EOSL model 3D-1.

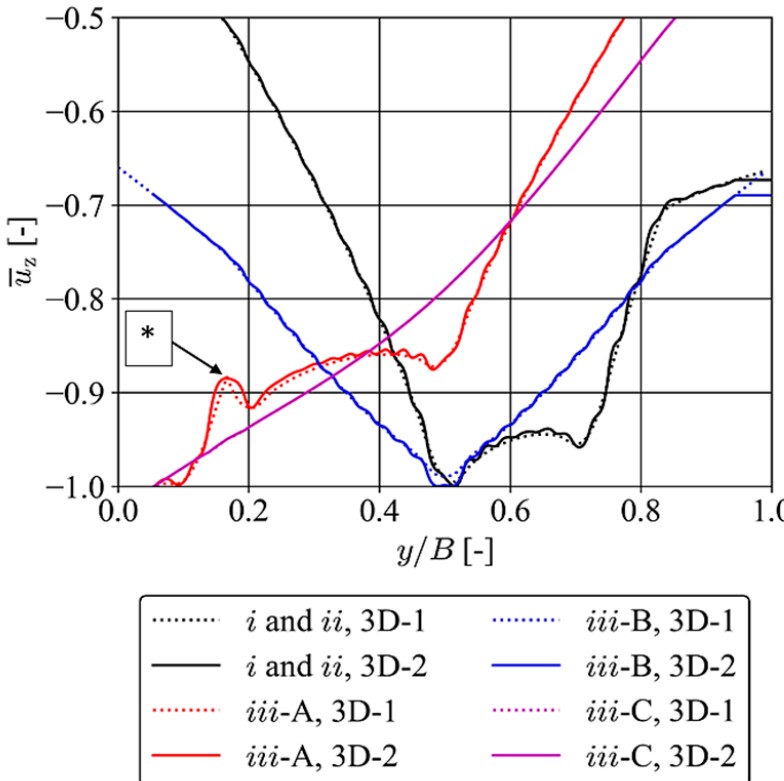

**Figure 14.** Vertical displacements for global models (3D-1) and level 2 sub-models (3D-2) for all five investigated regions. The zone * illustrates the effect of thick faces.

It can be seen from Figure 14 that there is a very good agreement between the global model 3D-1 and the level 2 sub-models 3D-2 in terms of the average displacements. Only the fluctuation of the free plate fields between two bottom core-to-face joints diverged, as was expected due to the incorporation of the 3D geometry in the model 3D-2. Furthermore, Figure 14 shows the shortcoming of the EOSL analysis with respect to thick-face effects and concentrated loads/supports (see Allen [33] and O'Connor [34]). This can be seen for the region iii-A at $y/B$, which was slightly less than 0.2 (see the mark * in Figure 14). The EOSL model (3D-1) showed a discontinuity in the deformation pattern related to the partial deformation with respect to transverse shear as an effect of the first-order shear deformation theory. In the level 2 sub-model (3D-2), this point rotation was prevented by the local bending stiffness of the bottom face. As can be seen in Figure 1, this had no effect on the average global response in this case. However, there might exist cases where this has an impact on the global average response.

Generally, it could be concluded from this figure that a simplified EOSL model can accurately capture average displacements in a CCSSP, which makes this fast analysis method efficient.

### 3.2. Impact of Weld-Region Deformations on Weld Stress Predictions

To evaluate the impact of the weld-region deformations on the nominal stresses in the welds, the model 2D-1 (Figure 4) with five different spring setups (*k*-setups) was evaluated, and the results were compared to those obtained from the model 2D-3 (Figure 6) as a reference. The five *k*-setups are given in Table 5 (where *calc* indicates the calculated value shown in Table 4) and refer to (1) rigid weld, (2) weld acting as a hinge, (3) incorporation of a rotational spring, (4) incorporation of rotational and *z*-translational spring, and (5) incorporation of rotational and both *y*- and *z*-translational springs.

**Table 5.** Spring stiffness setups.

| Setup | $k_{\phi 1}$ (kNm/rad) | $k_{\phi 2}$ (kNm/rad) | $k_{z1}$ ($10^6$ N/m) | $k_{z2}$ ($10^6$ N/m) | $k_{y1}$ ($10^6$ N/m) | $k_{y2}$ ($10^6$ N/m) |
|---|---|---|---|---|---|---|
| 1 | $\infty$ | $\infty$ | $\infty$ | $\infty$ | $\infty$ | $\infty$ |
| 2 | 0 | 0 | $\infty$ | $\infty$ | $\infty$ | $\infty$ |
| 3 | calc | calc | $\infty$ | $\infty$ | $\infty$ | $\infty$ |
| 4 | calc | calc | calc | calc | $\infty$ | $\infty$ |
| 5 | calc | calc | calc | calc | calc | calc |

Three different cells of the beam were investigated, as indicated in Figure 6. The three cases were subjected to: cell (1) panel-level actions $M_y$ and $Q_y$, cell (2) global actions $M_y$ and $Q_y$ together with DAL, and cell (3) subjected to panel-level bending $M_y$ alone. All four welds in the unit cells were investigated and numbered according to Figure 15a.

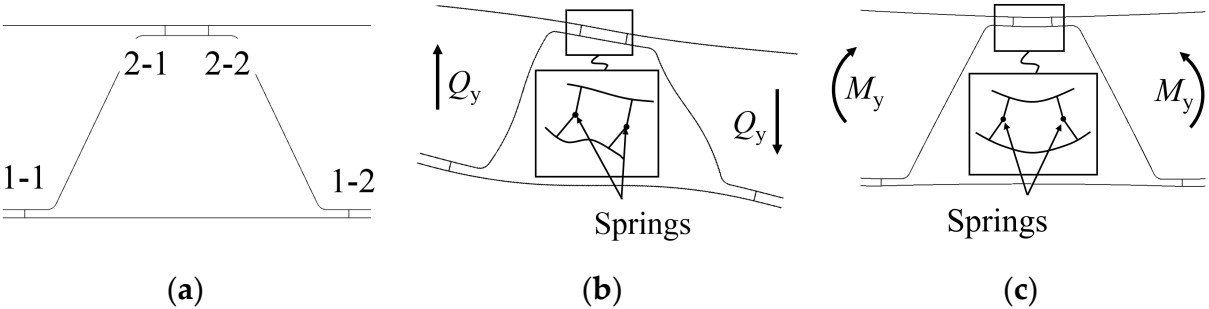

(a)  (b)  (c)

**Figure 15.** Corrugated-core steel sandwich unit cell: (**a**) weld numbering, (**b**) weld deformation due to panel-level shear, and (**c**) weld deformation due to panel-level bending moment.

Figure 15b,c show typical weld deformations from weak direction panel-level bending moment $M_y$ and shear force $Q_y$, respectively. As can be seen from Figure 15b,c, in one of the top face welds (weld 2-1), $Q_y$ and $M_y$ will interact, while in weld 2-2, the panel-level forces will cause weld deformations that counteract each other. The weld deformations from DAL add to these weld deformations, mainly in the top face. It should be noted that a perfect weld-region stress prediction is not reasonable to aim at by this simplistic approach including three degrees of freedom of a complex structural behaviour in a local region.

The results from the five different cases with different *k*-setups are shown in Table 6. A direct and undisputable conclusion from this investigation is that the weld-region rotational spring $k_\varphi$ is vital for accurately predicting the weld stress (compare *k*-setup 1, 2, and 3 in Table 6).

Comparing the *k*-setups 3 and 4, where the difference is that setup 4 includes weld deformations in the *z*-direction, it is clear that the vertical spring $k_z$ had no effect on the bending stresses originating from the panel-level bending action (see cell 3 results). However, $k_z$ had a minor effect on the weld stresses when the deformation of the panel originated from $Q_y$ (see cell 1 and compare *k*-setups 3 and 4). When investigating the effect of the shear spring $k_y$, the effect was directly opposite, and it had an effect on the weld bending stresses originating from panel-level bending. From all the results given in Table 6, it is shown that the rotational degree of freedom has the largest influence on the bending stresses in the welds, and including the effect of the translational degrees of freedom *y* and *z*, the stress predictions became more accurate in general, even though there exist two investigated welds where the error increased by an insignificant amount. It is noted that the maximum error of 9.2% was found in the weld 1-2 of cell 2, which had only 4% nominal bending stress compared to the maximum stressed weld, weld 2-1 of cell 2. In the analyses regarding the effect of the weld-region stiffness shown in this section, no effect on the other nominal stress components (from shear or axial force acting on the weld) was identified.

**Table 6.** Errors percentage associated with the frame model (2D-1) in comparison with the solid model (2D-3) as reference, regarding nominal bending stress prediction for different spring setups.

| k-Setup | Weld 1-1 | | | Weld 1-2 | | |
|---|---|---|---|---|---|---|
| | Cell 1 | Cell 2 | Cell 3 | Cell 1 | Cell 2 | Cell 3 |
| 1 (rigid) | 323.3 | 279.5 | 156.7 | 385.9 | 597.7 | 155.9 |
| 2 (hinge) | −100.0 | −100.0 | −100.0 | −100.0 | −100.0 | −100.0 |
| 3 ($k_\varphi$) | −0.7 | 1.4 | 7.5 | −3.5 | 59.3 | 7.2 |
| 4 ($k_\varphi$, $k_z$) | 0.0 | 2.0 | 7.5 | −2.5 | 57.0 | 7.2 |
| 5 ($k_\varphi$, $k_z$, $k_y$) | −1.1 | −1.7 | −1.9 | −0.4 | −9.3 | −2.2 |
| *k*-setup | Weld 2-1 | | | Weld 2-2 | | |
| | Cell 1 | Cell 2 | Cell 3 | Cell 1 | Cell 2 | Cell 3 |
| 1 (rigid) | 549.1 | 333.3 | 328.6 | 215.4 | 215.4 | 324.4 |
| 2 (hinge) | −100.0 | −100.0 | −100.0 | −100.0 | −100.0 | −100.0 |
| 3 ($k_\varphi$) | −5.4 | 0.9 | 13.1 | 5.4 | 5.4 | 12.0 |
| 4 ($k_\varphi$, $k_z$) | −1.3 | 1.7 | 13.1 | 4.2 | 4.2 | 12.0 |
| 5 ($k_\varphi$, $k_z$, $k_y$) | −3.8 | −1.3 | −2.0 | −0.2 | −0.2 | −2.9 |

### 3.3. Panel-Level Sectional Forces

This section investigates the capability of the global model 3D-1 with an EOSL deck in predicting the panel-level sectional forces. For this purpose, the results from model 3D-1 are evaluated and compared with those based on the model 3D-2 as a reference.

Figure 16 shows the orientation of the panel-level sectional forces. In order to calculate panel-level sectional forces in the 3D-2 model, the integration of stresses on a specific section was calculated and used. This was performed here using the built-in Abaqus-function free body cut (FBC); see Simulia [35]. The FBC utilises nodal forces and moments that are derived from the shell element stresses and summed onto the nodal points. All nodal forces within a cut are summed with respect to the neutral axis into panel-level sectional forces and moments. For cuts in the *xz*-plane, a length of two elements was used (40 mm), and for cuts in the *yz*-plane, a unit cell length of 2*p* was used; see Figure 16.

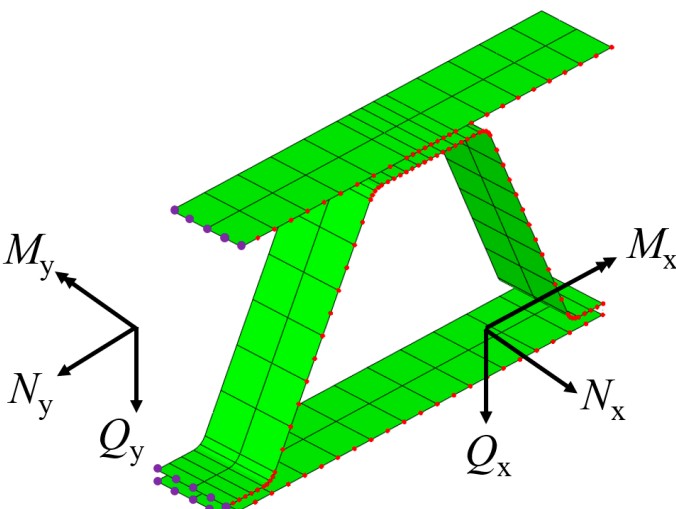

**Figure 16.** Panel-level sectional force directions and active nodes for calculation of panel-level sectional forces (based on free body cut); the red nodes are used for the yz-plane, and the purple nodes are used for the xz-plane.

The sectional forces for both models (3D-1 and 3D-2), loaded according to region i and ii, are shown in Figure 17a–d. In Figure 17a,b, the results for a section in the *yz*-plane when *x* is centric in the loaded span, i.e., output line 1 according to Figure 9, are shown. Figure 17c,d show results for a section in the *xz*-plane at *y* = *B*/2, i.e., output line 2 according

to Figure 9. In Figure 17, all sectional forces ($N_x$, $Q_x$, $M_x$, $N_y$, and $Q_y$) are normalised by their respective maximum value within the investigated output line. This is denoted by the bar over each notation. The notation $\overline{LE}$ indicates normalised load effect, i.e., $\overline{N}_x$, $\overline{M}_x$, $\overline{Q}_x$, $\overline{N}_y$, or $\overline{Q}_y$. The solid lines show results from 3D-1, whereas markers show result from the model 3D-2.

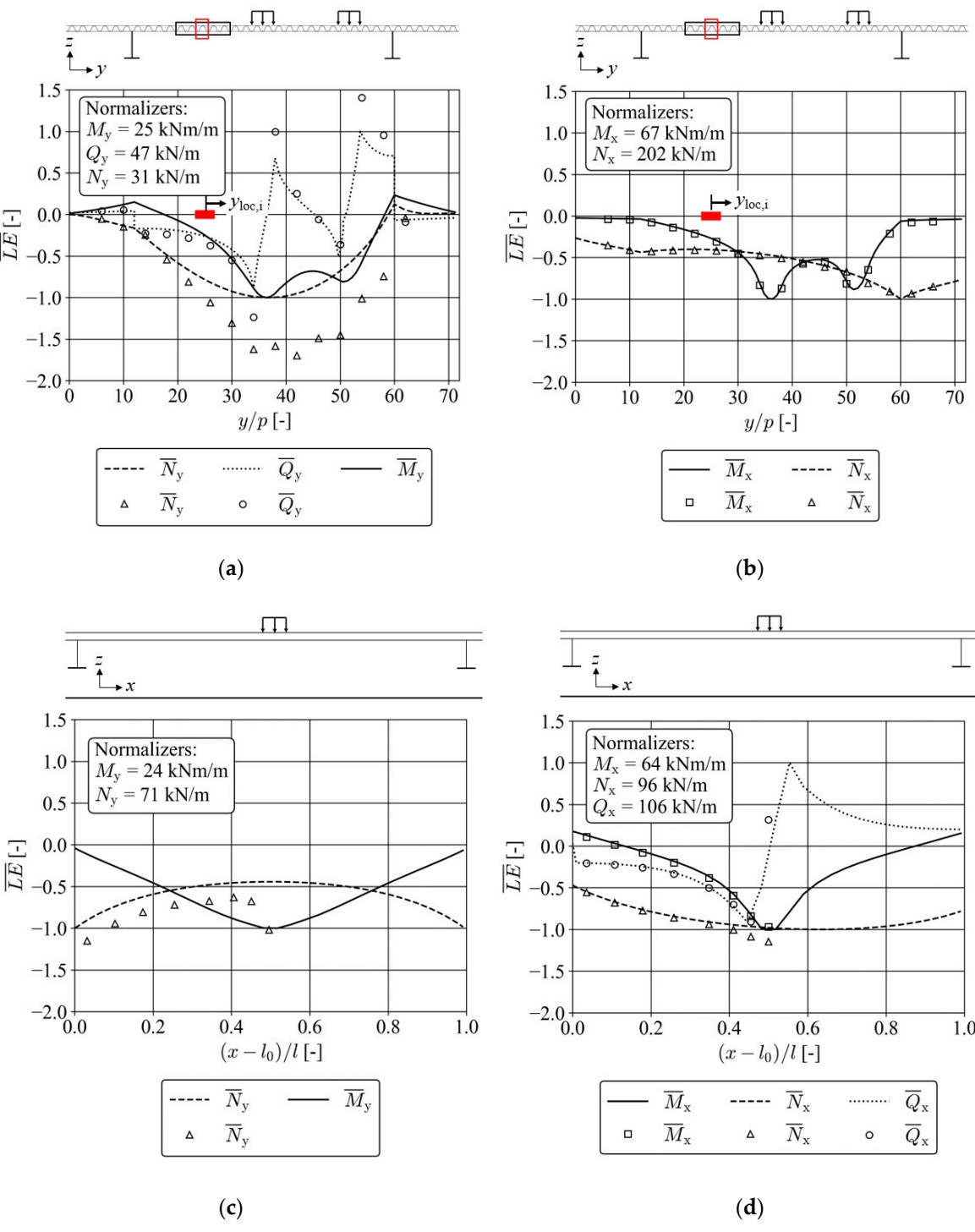

**Figure 17.** Panel-level sectional forces obtained from region i and ii load case for (**a**) load effects in the *y*-direction for output line 1, (**b**) in the *x*-direction for output line 1, (**c**) load effects in the *y*-direction for output line 2, and (**d**) in the *x*-direction for output line 2. Lines show results from 3D-1, and markers show results from 3D-2.

Figure 17b,d show that the EOSL model 3D-1 predicted the panel-level sectional forces in the stiff direction with high accuracy. The only discrepancy was found for locations directly under the applied load, which was expected. $M_y$ cannot be calculated in a precise manner using the FBC methodology, and it was thereby left outside of this study. Figure 17a,c show that the weak direction shear force $Q_y$ was also predicted well for locations away from supports and loads. However, a significant discrepancy was revealed concerning the weak direction membrane force $N_y$. At least two aspects contributed to the $N_y$ error. The first one stems from the fact that through-thickness compression was neglected in the EOSL model 3D-1. Under the localised patch loads, the panel was compressed in the thickness direction, and the panel core tended to expand in the horizontal plane (dominantly in the weak direction due to the lower stiffness). However, the core was constrained by surrounding structure and the webs of the main girders, yielding a membrane force. This effect is shown schematically for a constrained solid structure in Figure 18a.

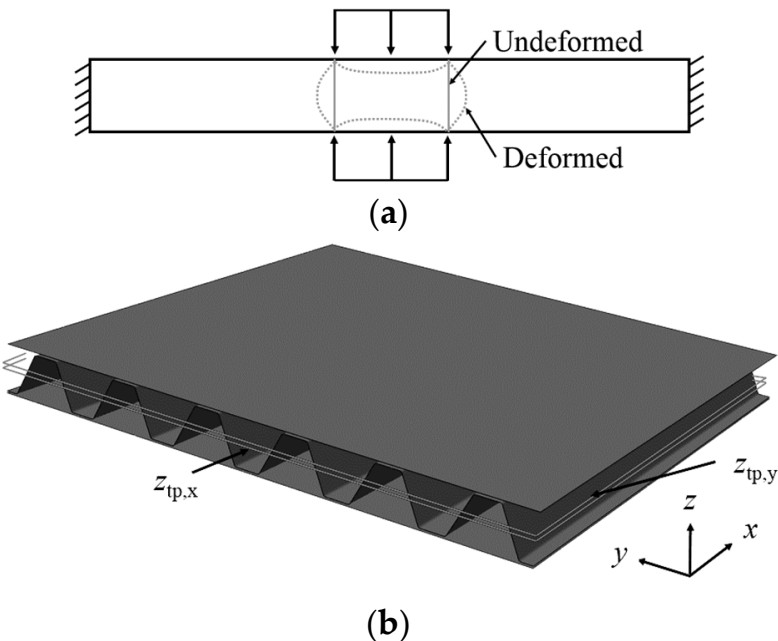

**Figure 18.** Sources of discrepancy in panel-level weak direction membrane force: (**a**) through thickness compression and (**b**) shape orthotropy.

The second source of error relates to the differences in the neutral layer that was different in the two different directions in this shape-orthotropic panel; see Figure 18b. The EOSL was modelled geometrically at the neutral plane with respect to the stiff direction; this yielded an offset with respect to the weak direction. Therefore, when bending of the transverse girder takes place, the weak direction membrane action in the deck is located at a *z*-position that is slightly different compared to the 3D model. This can be seen by the constant error in Figure 17c on the left side of the patch load. In a more general sense, this is related to the absence of a coupling matrix that relates bending and membrane panel-level actions, the so-called B-matrix (see Reddy [36], and Kok and Blaauwendraad [37]), which is assumed to be zero in the analyses of the model 3D-1. Thus, this has an effect both when bending the transverse girder and in the free span between the main and transverse girders.

### 3.4. Stress Predictions

In this section, the validity and accuracy level of the presented combined sub-modelling approach for predicting the stresses in the constituent core and face plates as well as the weld-region of CCSSP structures is verified and studied in comparison with a detailed 3D reference model.

### 3.4.1. Constituent Plates Stresses

Global Actions (Region i)

In the region where the panel is subjected to global actions alone (region i), $N_y$, $M_y$, and $Q_y$ caused normal stress in the $y$-direction in the constituent plates of the cross-section. The shear force $Q_y$ caused a bending moment $m_Q$, and the membrane force $N_y$ caused membrane force in the face plates, $n_N$; see Figure 19a,b, respectively. As shown in Nilsson [14], the weak direction bending moment $M_y$ caused both a local membrane force $n_M$ and a local bending moment $m_M$ in the face plates (Figure 19c).

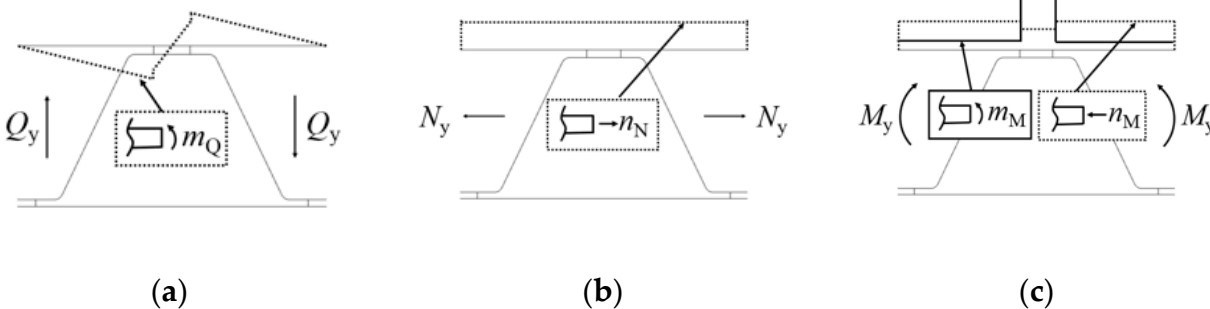

|     |     |     |
| --- | --- | --- |
| (**a**) | (**b**) | (**c**) |

**Figure 19.** Top face sectional forces originating from the panel-level sectional forces: (**a**) shear force, (**b**) normal force, and (**c**) bending moment.

A fundamental assumption of the EOSL approach is that the global sectional forces are constant over a unit cell, as shown in Figure 19. The panel-level sectional forces were captured in the EOSL model (3D-1) at the centre of the investigated cell (Figure 20).

For the EOSL approach calculations, the local stresses from $Q_y$ were calculated using the methodology given in Nilsson et al. [12]. The local membrane force $n_N$ from $N_y$ was directly calculated under the assumption that the face plates carry this force alone. Local bending moment $m_M$ and membrane force $n_M$ from $M_y$ were calculated using the model 2D-1 with spring setup 5 according to Table 6. From 2D-1, the local forces and moments in the face plates were captured at a cell centrically between the two loads, where a pure state of panel-level bending moment exists. The captured stresses were then multiplied by the valid panel-level sectional force from the EOSL model 3D-1 centrically in the investigated cell.

Figure 21a shows the normalised top and bottom face normal stress based on the described EOSL approach, $\sigma_{ESL}$, for a unit cell subjected to pure global action compared to the stresses in the 3D shell element model, 3D-3a. A local coordinate for the investigated cell is introduced: $y_{Reg\,i}$, where 0 is at $y = 25p$ and directed in the positive $y$-direction. Figure 21a-1,a-2 show all the different components that contribute to the total stress (denoted with indices according to Figure 19) in the EOSL approach. It is evident that $Q_y$ and the local membrane force $n_M$ from $M_y$ are the two major components.

It can be concluded from the curves shown in Figure 21 that there is a slight discrepancy between the two modelling approaches. One reason for this discrepancy is the assumption of constant sectional forces over the studied cell. For that reason, the use of an interpolation methodology was investigated. The top face-plate field S in Figure 20 is used here as an example to describe the interpolation methodology. An assumption was made that the local membrane force was constant in one face-plate field. For corrugated-core sandwich beams (e.g., Figure 4), this assumption was validated in [14]. Based on the EOSL approach with constant sectional forces, however, the magnitude for $M_y$ was captured at the point $P_1$; see Figure 20. In the adopted interpolation method, the average value of the magnitudes in the positions $P_1$ and $P_2$ was used instead. The shear force $Q_y$ caused a linearly distributed local bending moment $m_Q$ in the plate field S in Figure 20.

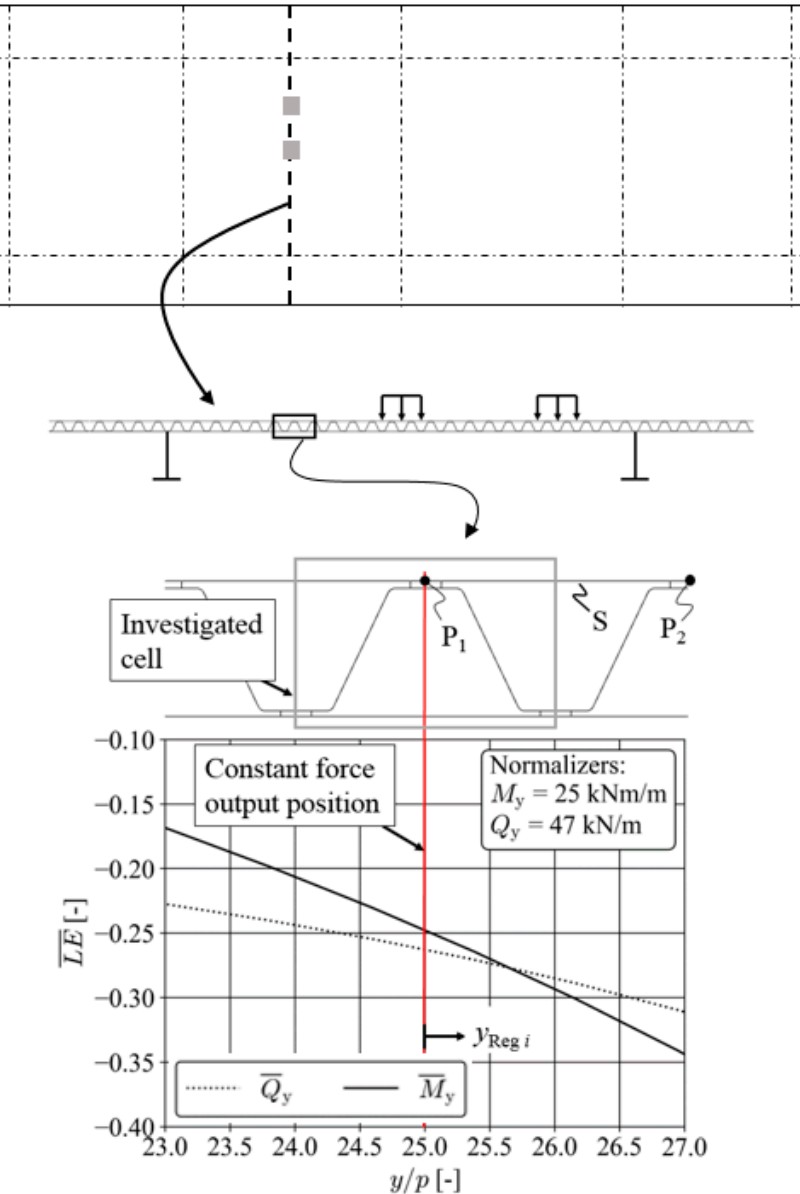

**Figure 20.** Position and panel–level sectional force distribution in region i.

In the interpolation methodology, $Q_y$ for the plate field S was calculated at the points $P_1$ and $P_2$, and the local bending moment was calculated at each weld position limiting the plate field, and a linear distribution was assumed within S. Figure 21b shows the top and bottom face surface normal stresses in the $y$-direction based on the EOSL approach including this interpolation, and the results were compared to those obtained from the 3D shell element model 3D-3a. A slight improvement can be seen compared to the results shown in Figure 21a. However, the accuracy can only be considered good enough for application in preliminary analyses of structures.

As it was shown, even after consideration of the non-constant sectional-force distribution over the unit-cell, there was still a discrepancy between the two investigated modelling approaches for the locations where panel-level sectional forces were acting alone. One observation from Figure 21a,b is that there was a discrepancy in the slope of the stress distribution in the different plate fields. This discrepancy can be explained by investigating the distribution of the face-plate membrane force. Using the EOSL approach, this distribution is assumed to be constant within one plate field.

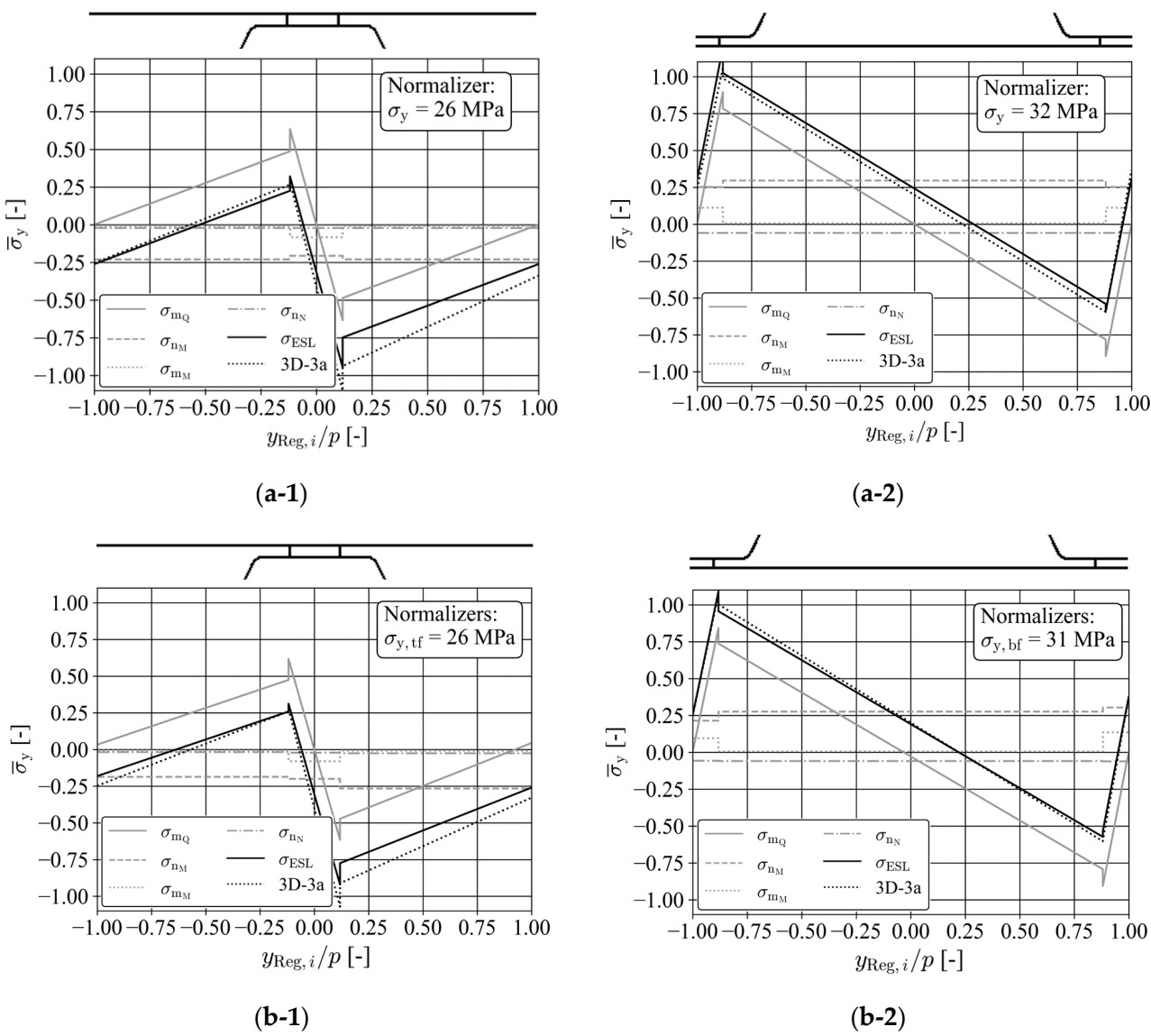

**Figure 21.** Comparison of stresses for the ESL approach assuming (**a**) "constant" panel–level sectional forces and (**b**) "interpolation" of panel-level sectional forces over a unit cell and the sub-modelling approach in a region subjected to panel-level sectional forces alone: (**1**) top face and (**2**) bottom face.

Figure 22 shows the $y$-direction top face membrane force $n_y$ of the model 3D-3a around the position $y_{\mathrm{Reg}\,i} = 0$. The normalised membrane force $\bar{n}_y = n_y/n_{y,\mathrm{max}}$, where $n_{y,\mathrm{max}}$ is the maximum membrane force in absolute values in the visualised area. The $n_y$ distribution in Figure 22a shows the same general trend as the main contributing factor to this force, $M_y$ (see Figure 17), with decreasing magnitudes with decreasing $y$-values due to load-spreading in the $x$-direction. However, within one top face-plate field (plate field S is again shown in Figure 22), the compression increases with decreasing $y$-values. This relates to the fact that, on this part, the panel-level bending force in the stiff direction, $M_x$, is also decreasing. This yields that the top face contractions of the weld lines in the $x$-direction at each side of the plate field S are divergent. These divergent contractions in the $x$-direction cause a constraint effect for the top face-plate fields that yields increasing $y$-direction compressive membrane force at the less-contracted weld. In the bottom face, this effect is opposite. Another reason for the discrepancy between the two approaches is that the interpolation methodology is just an approximation and does not describe the actual internal distribution of forces. The misprediction of the sectional forces shown in Figure 17 is also a source of error.

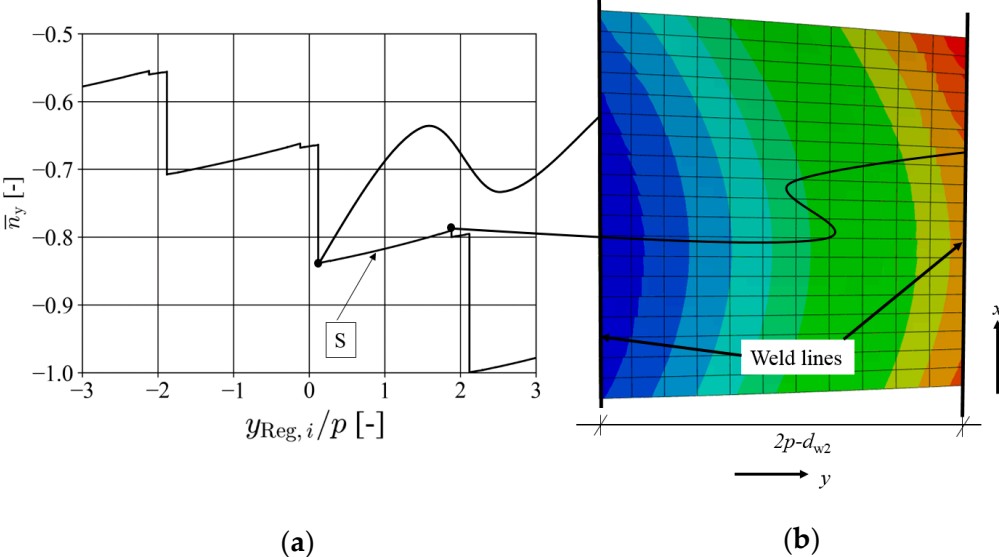

(**a**)
(**b**)

**Figure 22.** Distribution of weak direction membrane force along the *y*-direction obtained from 3D-3a (**a**) as a function of $y_{\mathrm{Loc}\,i}$ and (**b**) contour plot of plate field S.

In the above calculations, the model 3D-3a was used as a reference for the investigations obtained from the EOSL approach. In Figure 23, the face plate's outer-surface- and core plate's upper-surface stresses of 3D-3a are validated by comparison with the stresses in the solid element model 3D-3b. For the core plate, the investigated stress component was taken in its local coordinate system.

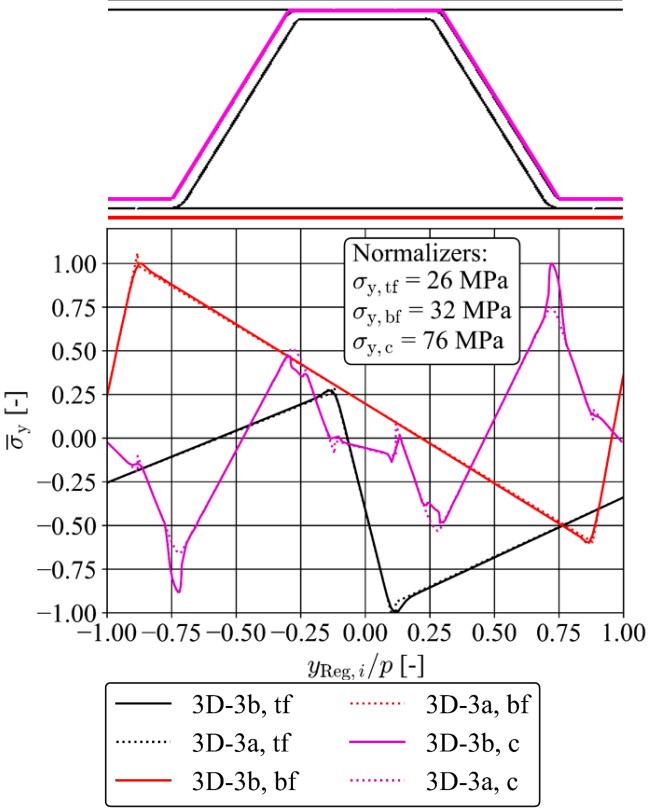

**Figure 23.** Normal stress distribution in the top and bottom face plates and the core plate on the surface of the plates obtained from the shell element model (3D-3a) and the solid element model (3D-3b) for region i.

As it can be seen in Figure 23, the shell element model can predict the stresses with a high accuracy. The only discrepancy in the face plates, which was small, was in the region close to the welds. This discrepancy is related to the discrete point connection between the core and face plates in the shell element model compared to the distribution of the weld that is included geometrically in the solid element model. For the core plate, there was also a discrepancy between the two models at the corner radius. This is due to the fact that the shell element model cannot capture the non-linear normal stress distribution through the thickness of the plate that occurs in curved sections. For this case, the discrepancy was significant due to the small ratio of radius over thickness ($R/t_c \approx 1.2$).

Region under DAL (Region ii)

In this section, the two approaches are considered again, this time in region *ii*, i.e., including the effect of DAL. The effect of DAL was calculated for the EOSL approach based on the model 2D-2 (see Figure 5) and super-positioned to the stresses from panel-level sectional forces that were calculated in the same manner as in region i, i.e., based on sectional forces only (see "Global Actions (Region i)" in Section 3.4.1). The location of region *ii* and the corresponding load case are shown in Figure 9b. The cell that was investigated in this section was located at $y = 35p$, and the relevant panel-level sectional forces can be seen in Figure 17. A local coordinate for the investigated cell is introduced: $y_{\text{Reg }ii}$, where its origin is at $y = 35p$ and directed in the positive $y$-direction.

Figure 24a,b show comparisons between the EOSL approach and the 3D shell element model 3D-3a in terms of normal stresses on the outer surface of the top- and bottom-face plates in the $y$-direction, respectively. As in the previous section, all stress components that compose the EOSL approach are shown separately.

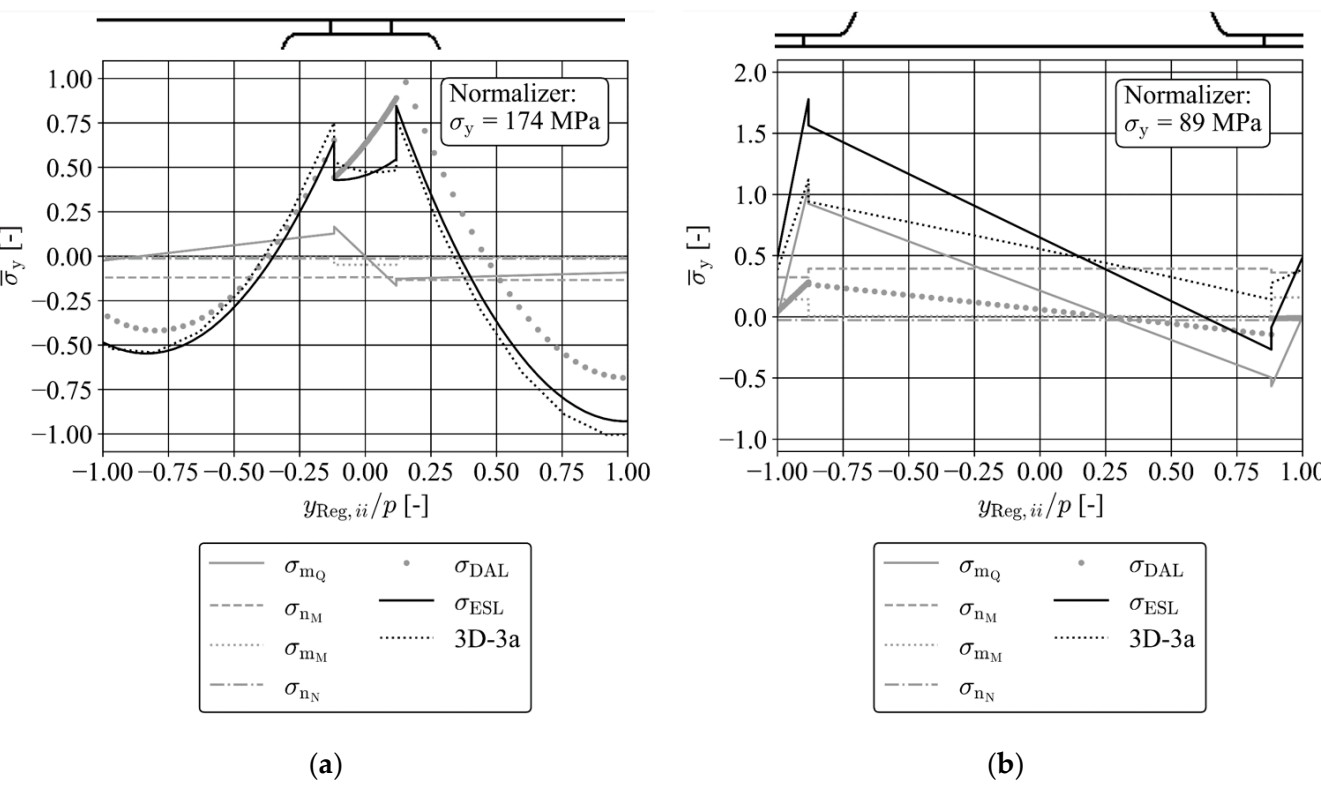

**(a)**                                                                                **(b)**

**Figure 24.** Comparison of stresses based on the ESL approach assuming interpolation of panel–level sectional forces over a unit cell and those obtained from the sub-modelling approach in a region subjected to panel-level sectional forces and directly applied load: (**a**) top face and (**b**) bottom face.

It is evident from Figure 24 that in the top face of this region, the DAL effects are dominant. The stress prediction in the top face had a minor discrepancy in comparison

to the reference model 3D-3a, and in this aspect, the EOSL approach can be seen as an approximation. However, Figure 24b shows a significant discrepancy with respect to the bottom face stresses. There are several reasons for the mispredictions in this region. One aspect is that the studied region is directly under a patch load, where the panel-level sectional forces are not expected to be correct in the EOSL model. One reason for this is that the EOSL model is, as was mentioned in Section 2, incapable of capturing the through-thickness effects in a sandwich panel. These effects are expected to be substantial in this region. Another aspect is the 2D nature of the simplistic DAL model 2D-2, which overlooks the "real" forces distribution from the top face—through the core—to the bottom face in the depth *x*-direction. The authors performed 3D analysis analogous to the load and BC setup as 2D-2. This analysis showed that this aspect only had a minor effect on the results, and it was not the major source for the misprediction. A third source of error was the bottom plate constraint conditions of the DAL model 2D-2; see Figure 24a.

Investigating the bottom face stress in Figure 24b and the corresponding deformed shape, it was evident that a major part of the applied load as transferred directly to the supports under the load. This yielded major local curvature close to the supporting points, which did not occur in the reference model 3D-3a. One way to "mitigate" this effect is to replace the rigid supports with springs, as shown in Figure 25.

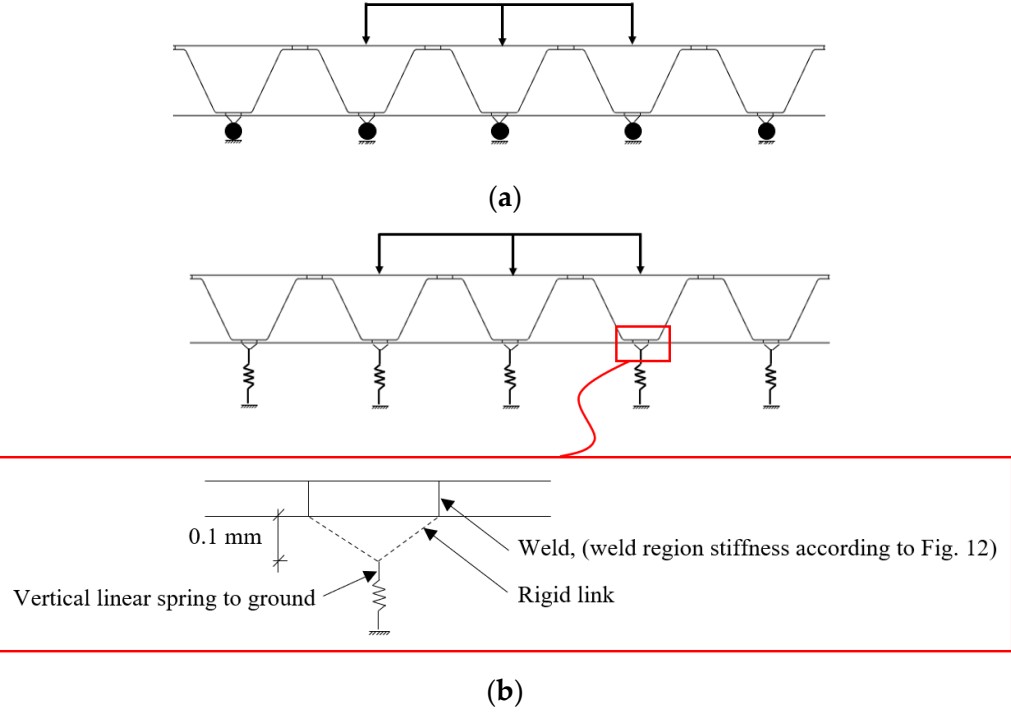

**Figure 25.** Model setups for analysis of the effect of directly applied load (DAL): (**a**) rigid supports and (**b**) spring supports (weld region stiffness according to Figure 12).

The spring model shown in Figure 25b provides a solution with an accuracy that is still below what can be tolerated even for approximate estimations. A direct conclusion of the work shown in this section is that the system is highly coupled; i.e., the local response at the region in the vicinity of the load is coupled to the global response, and an accurate decoupled model for the DAL is hard to provide. Another conclusion that can be drawn by investigating the results from the two models—at the position of the welds—is that the EOSL approach cannot be used to predict the weld stresses with high accuracy.

Figure 26 shows the face-plate outer surface and core plate upper surface stress comparison between the 3D shell element model 3D-3a and the 3D solid element model 3D-3b in region ii, directly under a patch load. It can be seen that the two models are in good agreement.

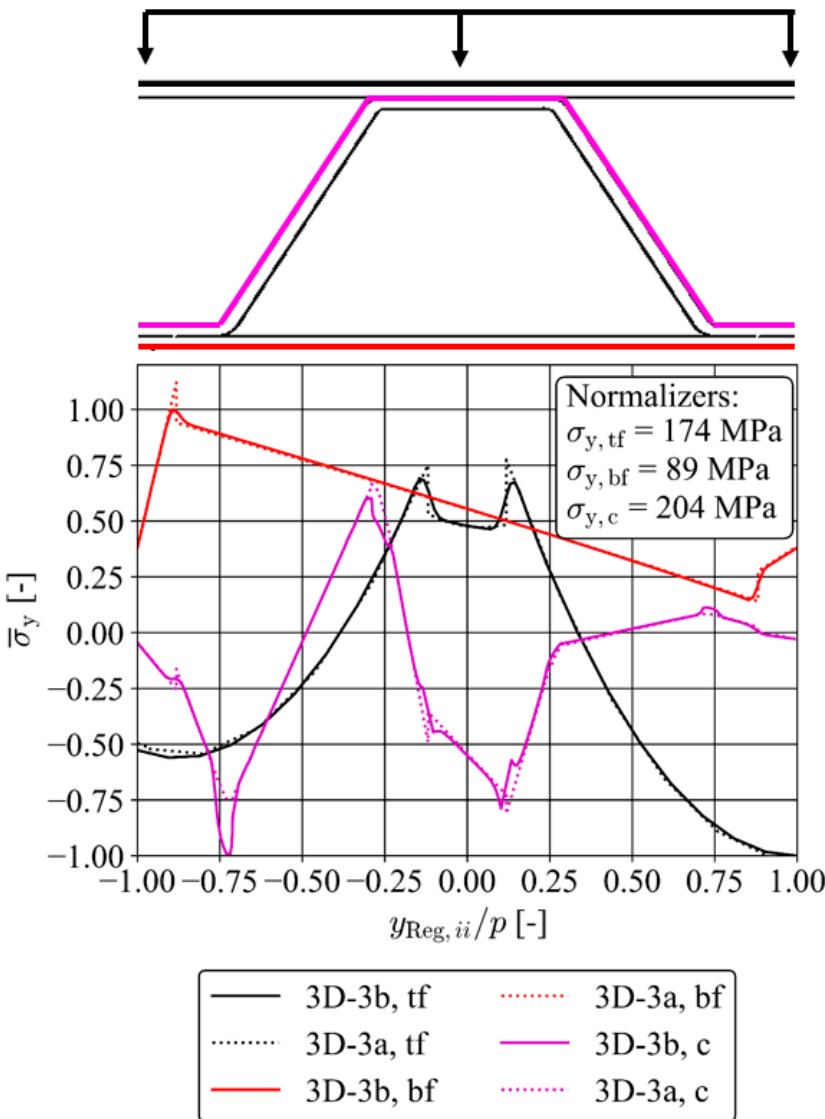

**Figure 26.** Normal stress distribution in the top and bottom face plates and the core plate on the surface of the plates, obtained from the shell model (3D-3a) and the solid model (3D-3b) for region ii.

It was also noted that the stress magnitudes were approximately two times larger in the top face than in the bottom. Another observation from Figure 26 is that the core stresses were high, and the maximum stressed point was in the corner radius, where there was a discrepancy between the two models. In a design situation, this needs to be considered.

Local Supports (Region iii)

Here, three various regions are investigated to cover different possibilities of the local transverse support effects on the CCSSPs; regions iii-A, B, and C, as shown in Figure 9. As the deficiencies of the EOSL approach for region ii have been shown, this approach was not expected to capture the structural behaviour well in these three more complex regions, which included highly localised load applications. Therefore, the approach was left outside of the following investigations. The main aim of the analyses in this section is to validate the sub-modelling approach, i.e., validating the model 3D-3a (shell) with 3D-3b (solid) as a reference, focusing on regions close to the supporting structure. Figure 27 shows comparative stress distributions for the two models for region *iii*-A, B, and C (for orientation, see Figures 8c(3)–(5) and 9c–e.

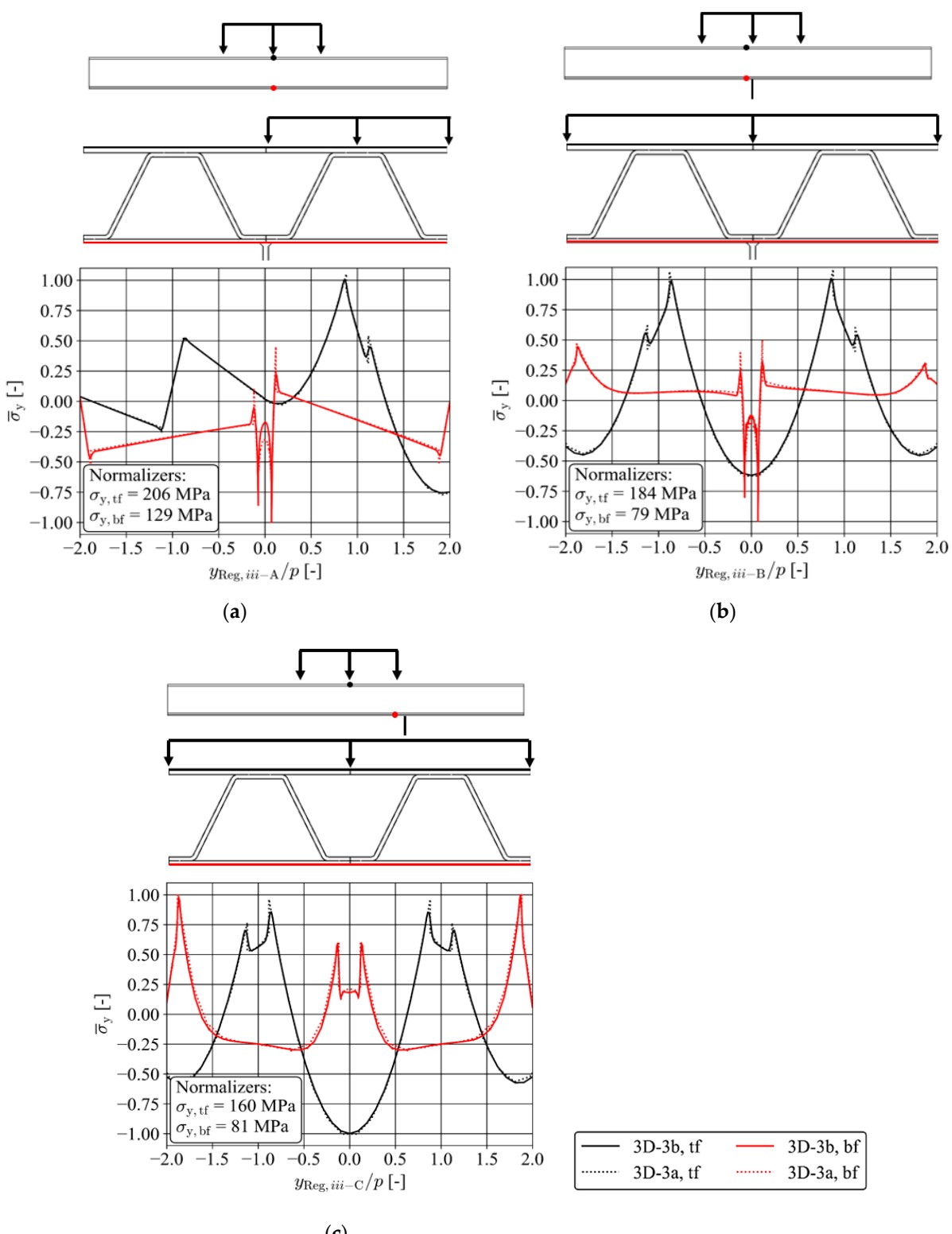

**Figure 27.** Comparative stress plots for 3D-3a (shell) and 3D-3b (solid) sub-models considering outer surface normal stress in the *y*-direction: (**a**) region iii–A, (**b**) region iii–B, and (**c**) region iii–C.

It can be seen from Figure 27 that also for positions close to supporting structures, the shell element models can predict stresses with high accuracy when the solid element model is considered as a reference model.

The only discrepancies were found at the connection points between the core and the face plates in the shell element models. Region iii-A as created in order to reflect a harsh situation for the core-to-face joint at the connection to the main girder. The bottom face stresses had their maximum compressive magnitude at the toe of the deck-to-main girder weld. However, it can be observed that the top face stresses were approximately 60% higher than in the bottom face plates in this region.

### 3.4.2. Weld Stresses

For all five investigated regions, the four centric top and bottom welds of the level 3 sub-models (3D-3a and b) were investigated. Even though there were discrepancies, the general conclusion is that the shell element model 3D-3a can predict all nominal stress components with an adequate accuracy for design purposes when the solid model 3D-3b is used as the reference for the studied case. In the following, selected examples of the results are presented and discussed.

### Global Actions (Region i)

Regarding region i, the curves of each stress component distribution (in the *x*-direction) were similar for all investigated welds. However, the magnitudes were slightly different. In this study, three nominal weld stress components were investigated: shear stress parallel to the weld line $\tau_{xz}$, shear stress perpendicular to the weld line $\tau_{yz}$, and the normal stress $\sigma_z$. The normal stress was investigated separately depending on its driving action, i.e., bending moment, i.e., $\sigma_{zM}$, or axial force, i.e., $\sigma_{zN}$. Figure 28a shows the four stress components for a bottom core-to-face weld in region i. In this region, there was no significant difference between the top and bottom face weld stress magnitudes, as the deck was subjected to panel-level sectional forces only. It can be seen that the accuracy of the results is high and that the stress levels are low. Low magnitudes for $\tau_{xz}$ are expected in this region; see $Q_x$ distribution in Figure 17. The normal stress from bending action in the weld, $\sigma_{zM}$, mainly originated from panel-level transverse shear force, $Q_y$, and to some extent from $M_y$. It is noticeable that the maximum and minimum shear force regions were evidently highly local at the exterior boundaries of the patch loads and spread widely as the loads were transmitted to the supporting structure, as also seen from the distributions given in Figure 17a. This is as a result of the high level of orthotropy of the panel and plays an important role interaction of multiple patch loads within a vehicle.

### Region under DAL (Region ii)

Figure 28b shows nominal shear and normal stresses in the top and bottom welds along the weld lines under the DAL (Region i). Also, for regions in the vicinity of the DAL, the shell element model 3D-3a predicted the stresses with high accuracy when compared to the more detailed solid model 3D-3b. The top weld was dominated by the normal stress originating from the local bending of the top face plate (and thus the weld).

It was observed, when compared to the weld in region i, that the DAL did not have a significant effect on the load effects in the bottom welds. Comparing the top weld stresses in region i (Figure 28a) and region ii (Figure 28b) revealed that there was approximately a factor as big as 30 times difference between the weld stresses. Considering the $Q_y$ distribution, it can be concluded that even if this panel is subjected to full design loads, the top weld lines under the patch loads are likely decisive rather than weld positions where the panel is subjected to global load effects alone.

### Local Supports (Region iii)

Figure 29a shows nominal shear and normal stresses in the top and bottom welds along the weld lines (*x*-direction) in region iii-A, which is located close to the main girder. The top core-to-face weld stresses were similar to those in region ii, considering both magnitude and distribution. Thus, the stresses were dominated by the local flexure originating from the DAL, with only a modest contribution from $Q_y$.

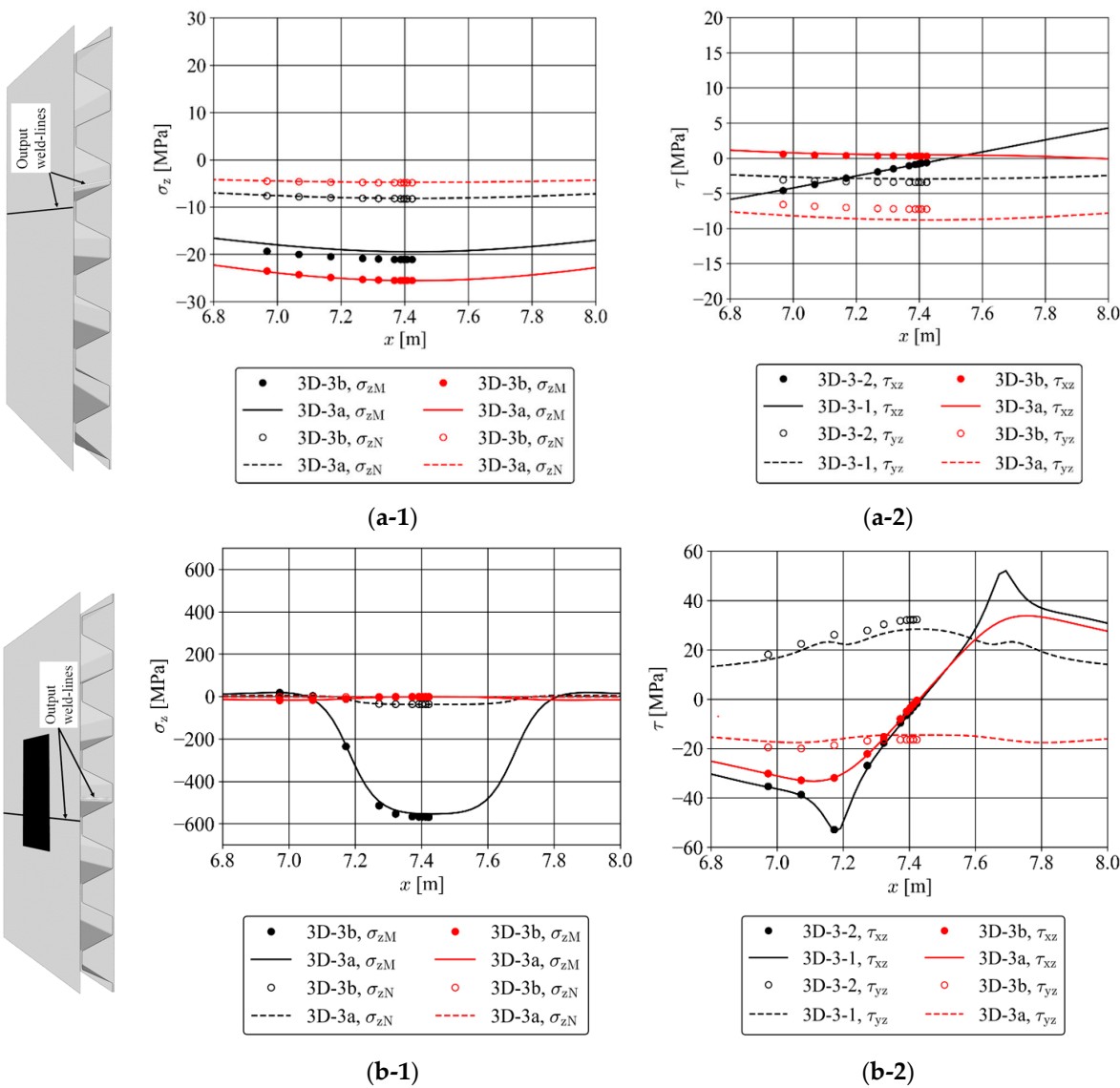

**Figure 28.** Distribution of the nominal weld stress along the weld line based on the shell model 3D-3a and solid model 3D-3b in: (**a**) region i: 1—normal stresses from bending and axial force, 2—shear stress components; (**b**) region ii: 1—normal stresses from bending and axial force, 2—shear stress components. (Solid black region indicates the surface of applied load.).

In fact, the situation was similar in all three region iii cases for the top welds. The bottom weld that was located close to the supporting structure is, however, distinctly different compared to region i and ii both from the perspective of distribution and magnitude. Higher shear stress perpendicular to the weld and normal stress from both bending and axial force were observed here. But even though this load case was created to yield a harsh situation for the welds at the supporting structure, the stresses were still only about 1/4 of those in the top welds. All other investigated bottom welds in this region (further away from the main girder web) had lower stress magnitudes. A modelling aspect that plays an important role in the distribution of stress in the core-to-face joints for the shell model 3D-3a is the connection configuration between the supporting structure and the bottom face of the deck. In the solid model, the weld throat was included in the geometry, as shown in Figure 30a. For the shell model, two principally different ways of modelling this connection are possible. The first type is a discrete point connection that couples the top node of the web of the supporting girder to the corresponding node in the bottom face of the panel; see Figure 30b. All six degrees of freedom are connected here. Alternatively, a

distributed interaction can be adopted, where coupling *C* contains all three translational degrees of freedom, and the rest of the nodes within a specific region *d* are coupled only with respect to translations in the *z*-direction. Here, the distribution region is defined by $d = t_w + 2l_{weld}$, where $l_{weld}$ is the leg length of the throat weld. Figure 29a shows the normal stress from bending of the weld for the two different approaches. It is shown that a distributed coupling yields results that are in agreement with the solid element model. The reason why the magnitude of the bending stress is larger in this case is that the distributed coupling keeps the bottom face straight, whilst the core is free to deform between the two weld lines. If the connection is performed in a discrete point, the bottom face is also deformed, causing less relative rotation of the two plates (and thus bending in the welds).

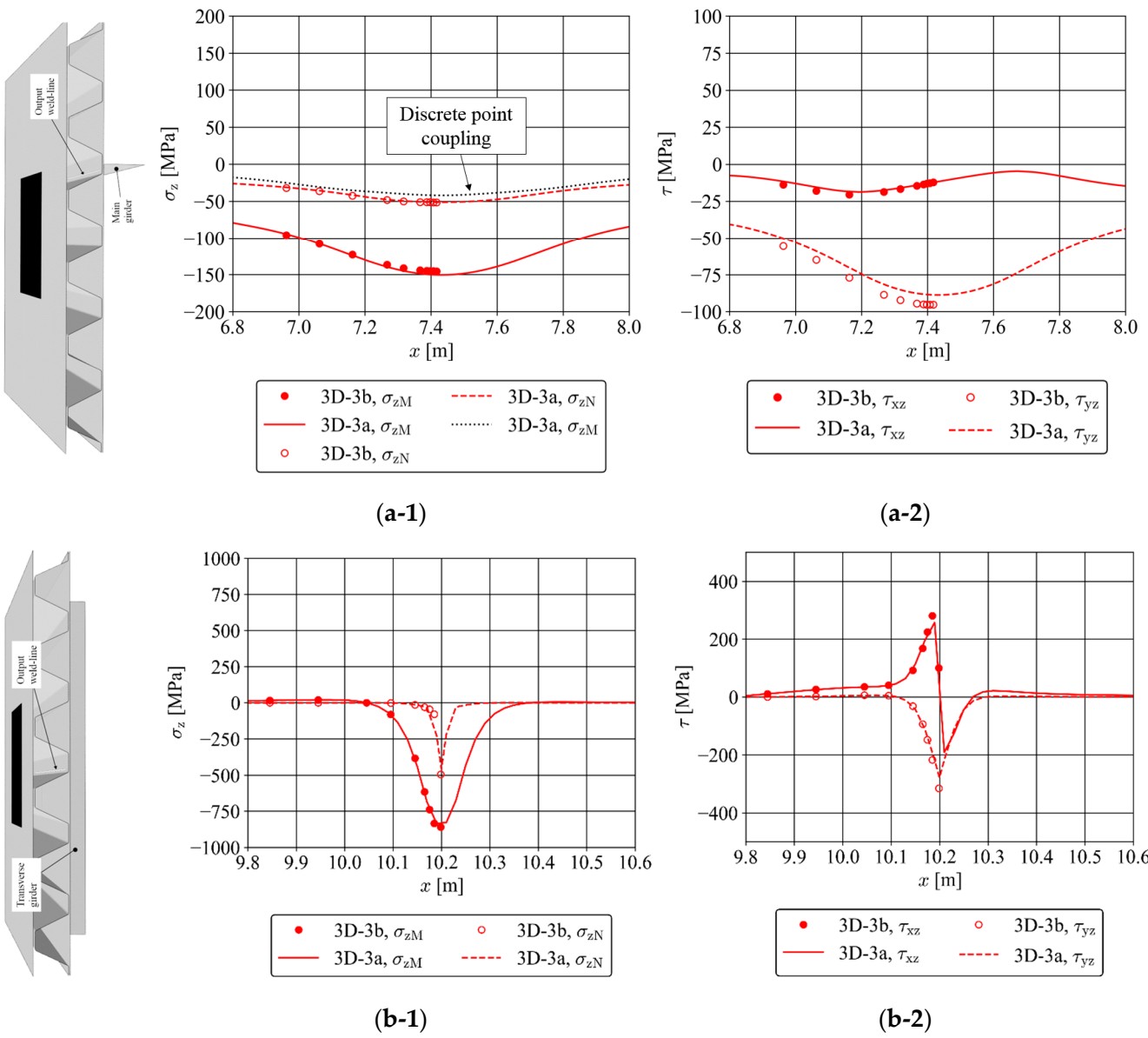

**Figure 29.** Distribution of the nominal weld stress along the weld line based on the shell model 3D-3a and solid model 3D-3b in: (**a**) region iii-A: 1—normal stresses from bending and axial force, 2—shear stress components; (**b**) region iii-C: 1—normal stresses from bending and axial force, 2—shear stress components. (Solid black region indicates the surface of applied load.).

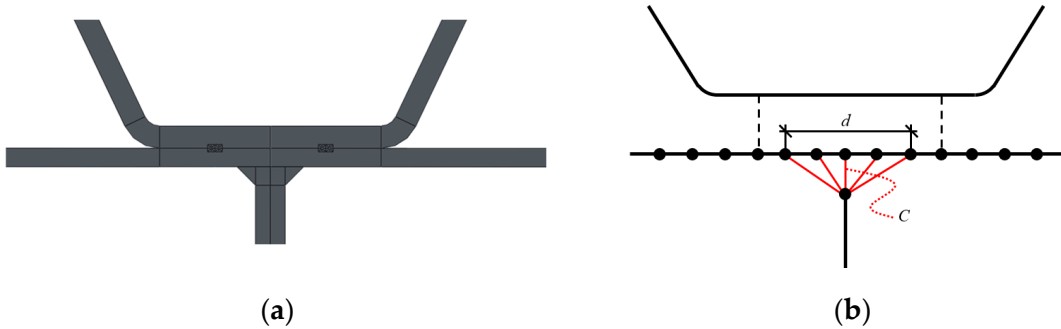

**Figure 30.** Geometric configuration of the modelled panel-to-web connection: (**a**) solid element model and (**b**) shell element model.

Figure 29b-1,b-2 shows the shear and normal stresses along the bottom face weld line in region iii-C, which was located close to the transverse girder; see full load-situation in Figure 9. The transverse support is perpendicular to the weld lines and core fold. The decisive stress components in this region were similar to those in region iii-B but of higher magnitudes. Also, for this case, it is shown that the shell model 3D-3a performed with high accuracy. Figure 29b shows that the bottom weld stresses, where the deck connects to a girder web that is perpendicular to the corrugation, are high. The normal stress from bending of the weld was more than five times higher than at the connection to the main support. The origin of these high stresses can be attributed to two different sources; first, the local patch load causes a compressive normal force in the welds that is high. Secondly, the bottom face is constrained to the transverse support, causing a stiff constraint line between the web of the transverse support and the bottom face, leading to high bending action in the core at this location.

## 4. Concluding Remarks

Production of steel sandwich panels has been recently enabled with the aid of laser-welding technology, which evidently offers great structural benefits in different areas of engineering applications from civil infrastructure to maritime technology. This paper, therefore, dealt with introducing an efficient combined modelling approach for gaining detailed structural response prediction for an accurate and reliable design and maximum utilisation of corrugated-core steel sandwich panel (CCSSP) lightweight structures. The capability and accuracy level of different modelling approaches for capturing structural displacements and stresses within CCSSPs, specifically in the weld region, on both local and global scales were examined and comparatively studied. The effect of applied local loads as well as different transverse supporting scenarios of the structural response of CCSSPs and the capability of different modelling approaches were investigated and discussed. It was demonstrated that the introduced global model, which uses an equivalent orthotropic single layer (EOSL) for the panel, predicts average deformations with a high accuracy. The results also showed that the mentioned modelling approach can be used to predict the face-plate stresses with a level of accuracy that could be used for approximate analysis in regions away from applied loads and the supports. This was also shown to be valid regarding the top face stresses in regions in the vicinity of applied loads. However, the mentioned modelling approach was found incapable of providing accurate estimation of load effects in the bottom plate in regions under the transverse local load, the so-called "directly applied load" (DAL).

The ability of the EOSL-based global model for prediction of panel-level sectional forces of the CCSSPs was also investigated. It was demonstrated that the EOSL-based global modelling approach was accurate considering forces in the stiff direction of the panel. Relatively good accuracy and agreement with detailed 3D reference models was also found with regard to the shear forces in the weak direction of the sandwich panel. The membrane force in the weak direction obtained from the EOSL global model was, however,

found less accurate when compared to the reference model. Two possible sources for this misprediction were identified: through-thickness compression and the shape-orthotropy of the panel. However, the results also showed that the contribution from the membrane force to the stresses in the face plates was modest.

The investigated sub-modelling approach using shell constituent configuration was found to exhibit high accuracy in conjunction with a more detailed 3D solid sub-model. This is valid both with respect to local load effects in the constituent plates as well as the weld region (panel-level sectional forces alone in the vicinity of DAL or the supporting area). In order for the sub-model approach to accurately predict structural detailed response, the shell model needs to incorporate the deformability of the weld region, which in this investigation was carried out via linear spring modelling. This implies the validity of the 2D approach derived in [26] as performed also in 3D. It was shown that the weld-region deformations in the vertical direction and the direction perpendicular to the weld line also have a minor effect on the state of stress in the welds.

**Author Contributions:** Conceptualization, S.R.A. and M.A.-E.; methodology, P.N.; software, P.N.; validation, P.N., S.R.A. and M.A.-E.; formal analysis, P.N.; investigation, P.N., S.R.A. and M.A.-E.; resources, P.N., S.R.A. and M.A.-E.; data curation, P.N.; writing—original draft preparation, P.N. and S.R.A.; writing—review and editing, P.N., S.R.A. and M.A.-E.; visualization, P.N.; supervision, S.R.A. and M.A.-E.; project administration, M.A.-E. and S.R.A.; funding acquisition, M.A.-E. and S.R.A. All authors have read and agreed to the published version of the manuscript.

**Funding:** This research was funded by the Norwegian Public Road Administration and the Swedish Transport Administration.

**Data Availability Statement:** Not applicable.

**Acknowledgments:** The work presented here has been funded by the Norwegian Public Road Administration and the Swedish Transport Administration, whose support is gratefully acknowledged. The numerical analyses were performed on resources provided by Chalmers Centre for Computational Science and Engineering (C3SE).

**Conflicts of Interest:** The authors declare no conflict of interest.

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
