# Peer review of "Laser-Welded Corrugated-Core Sandwich Composition—Numerical Modelling Strategy for Structural Analysis"

_jcs, doi:10.3390/jcs7090349_

Round 1

Reviewer 1 Report

In this paper, the authors introduced an efficient combined modelling method to product the structural response of corrugated-core steel sandwich panel (CCSSP) lightweight structures. The authors predicted CCSSP panel-level sectional forces using an EOSL-based global model, which is found to have good accuracy based on the comparation to a detailed 3D reference model, and analyzed the reason why the EOSL global model produces errors in the prediction of the membrane force in the weak direction. Besides, the authors discovered that the sub-modeling approach using shell composition configurations combined with a detailed 3D solid sub-model exhibited high accuracy. This research is comprehensive and detailed, and has important implications for achieving an accurate and reliable design and maximizing the use of CCSSP lightweight construction. Here are some things that could be improved.

Point 1. The full-text structure can be added in the introduction, so that readers can quickly understand the article.

Point 2. In the introduction, the author introduced the application prospects and difficulties of structural analysis of steel sandwich panels with corrugated core. Some relevant literatures can be added on this basis, so that readers can easily understand the research status of this structure.

Minor editing of English language required.

Author Response

Dear Anonymous Reviewer # 1

Thank you for reviewing our manuscript and providing valuable comments. The manuscript was updated based on your comments:

Reviewer: The full-text structure can be added in the introduction, so that readers can quickly understand the article

Authors: It was done based on the reviewer's comment.

Reviewer: In the introduction, the author introduced the application prospects and difficulties of structural analysis of steel sandwich panels with corrugated core. Some relevant literatures can be added on this basis, so that readers can easily understand the research status of this structure.

Authors: The Introduction section was reorganized, and relevant literature was addressed according to the reviewer’s comment.

Reviewer: Minor editing of English language required.

Authors: The entire manuscript underwent thorough proofreading and editing, and some minor amendments were made.

Yours sincerely,
S.R. Atashipour, PhD

Department of Mechanical Engineering
NVH & Experimental Mechanics Research Group
Kettering University
1700 University Ave., Flint, Michigan 48504, USA
Tel: +1 (810) 268-2060
[email protected]

Division of Dynamics
Department of Mechanics & Maritime Sciences (M2)
Chalmers University of Technology
SE-412 96 Gothenburg, Sweden
[email protected]

Reviewer 2 Report

This is a well written article regarding calculation of corrugated sandwich plates by laser welding.

Paper is acceptable for publication in its present form with some minor updates:

Page 1 Line 22 -26: Please review the type of font.

Figure 4: Please state briefly description for each variable and schematize the thickness of welding as well as the gap between plates and core.

Line 379 - 383: Please review alignment for table 1. 

Line 395-398: Please check alignment for table 2.

No additional comments to the Editor

Author Response

Dear Anonymous Reviewer # 2

Thank you for reviewing our manuscript and making valuable comments. Your comments were utilized and applied into the manuscript. The authors would like to briefly report what they changed in the manuscript on the basis of your comments as follow:

Reviewer: Page 1 Line 22 -26: Please review the type of font.

Authors: It was done.

Reviewer: Figure 4: Please state briefly description for each variable and schematize the thickness of welding as well as the gap between plates and core.

Authors: The figure is modified according to the reviewer’s comment, and a brief description of all the variables was added to the text.

Reviewer: Line 379 - 383: Please review alignment for table 1.

Authors: It was done.

Reviewer: Line 395-398: Please check alignment for table 2.

Authors: It was done.

Yours sincerely,
S.R. Atashipour, PhD

Department of Mechanical Engineering
NVH & Experimental Mechanics Research Group
Kettering University
1700 University Ave., Flint, Michigan 48504, USA
Tel: +1 (810) 268-2060
[email protected]

Division of Dynamics
Department of Mechanics & Maritime Sciences (M2)
Chalmers University of Technology
SE-412 96 Gothenburg, Sweden
[email protected]

Reviewer 3 Report

This paper is good description and I suggest it can be accepted in this Journal, my comments are below:

1, The abstract is too long, please write the main contents.

2, Why using EOSL model? Show the merit of reason.

3, In Fig. 25, please show the detail of spring supports.

The description is good.

Author Response

Dear Anonymous Reviewer # 3

Thank you for reviewing our manuscript and making valuable comments. The authors would like to briefly report what they changed in the manuscript on the basis of your comments:

Reviewer: The abstract is too long, please write the main contents.

Authors: The abstract was rephrased briefly to encapsulate the main contents according to the reviewer's comment.

Reviewer: Why using EOSL model? Show the merit of reason.

Authors: As part of the introduced combined sub-modeling technique, EOSL is used to efficiently obtain the displacements rapidly with relatively minimal computational effort, as highlighted in the manuscript. However, as the approach is incapable of capturing the stresses in the panel elements, the EOSL displacement outcome will be imported into the proposed sub-models, resulting in accurate stress predictions in the panel constituent plates. The description has been provided in the manuscript.

Reviewer: In Fig. 25, please show the detail of spring supports.

Authors: It was done based on the reviewer's comment.

Yours sincerely,
S.R. Atashipour, PhD

Department of Mechanical Engineering
NVH & Experimental Mechanics Research Group
Kettering University
1700 University Ave., Flint, Michigan 48504, USA
Tel: +1 (810) 268-2060
[email protected]

Division of Dynamics
Department of Mechanics & Maritime Sciences (M2)
Chalmers University of Technology
SE-412 96 Gothenburg, Sweden
[email protected]

Round 2

Reviewer 1 Report

Overall, the reviewer believes that the manuscript is suitable for publication in the J. Compos. Sci. for the subject therein treated.

Reviewer 2 Report

Good improvement of the article.

Reviewer 3 Report

This manuscript can be accepted in this Journal.